# Phosphatidylserine-exposing extracellular vesicles in body fluids are an innate defence against apoptotic mimicry viral pathogens

Rüdiger Groß [1], Hanna Reßin [1], Pascal von Maltitz [1], Dan Albers [1], Laura Schneider [2], Hanna Bley[2], Markus Hoffmann [3,4], Mirko Cortese[5], Dhanu Gupta [6,7], Miriam Deniz[8], Jae-Yeon Choi[9], Jenny Jansen[10], Christian Preußer[11], Kai Seehafer[12], Stefan Pöhlmann [3,4], Dennis R. Voelker[9], Christine Goffinet [10,13], Elke Pogge-von Strandmann[11], Uwe Bunz[12], Ralf Bartenschlager [5], Samir El Andaloussi [6], Konstantin M. J. Sparrer [1], Eva Herker [2], Stephan Becker[2], Frank Kirchhoff [1], Jan Münch [1] & Janis A. Müller [1,2] ✉

Some viruses are rarely transmitted orally or sexually despite their presence in saliva, breast milk, or semen. We previously identified that extracellular vesicles (EVs) in semen and saliva inhibit Zika virus infection. However, the antiviral spectrum and underlying mechanism remained unclear. Here we applied lipidomics and flow cytometry to show that these EVs expose phosphatidylserine (PS). By blocking PS receptors, targeted by Zika virus in the process of apoptotic mimicry, they interfere with viral attachment and entry. Consequently, physiological concentrations of EVs applied in vitro efficiently inhibited infection by apoptotic mimicry dengue, West Nile, Chikungunya, Ebola and vesicular stomatitis viruses, but not severe acute respiratory syndrome coronavirus 2, human immunodeficiency virus 1, hepatitis C virus and herpesviruses that use other entry receptors. Our results identify the role of PS-rich EVs in body fluids in innate defence against infection via viral apoptotic mimicries, explaining why these viruses are primarily transmitted via PS-EV-deficient blood or blood-ingesting arthropods rather than direct human-to-human contact.

Pathogenic viruses spread through body fluids and excretions. For instance, human immunodeficiency virus 1 (HIV-1) is mainly transmitted via semen[1], severe acute respiratory syndrome coronavirus 2 (SARS-CoV-2) via aerosols[2], Ebola virus (EBOV) via various body fluids[3], and Zika virus (ZIKV) and dengue virus via blood-feeding mosquito vectors[4]. For effective transmission, the virus must be shed into the relevant body fluid and encounter target cells at the exposure site. Additionally, viruses must evade innate defence

[1]Institute of Molecular Virology, Ulm University Medical Center, Ulm, Germany. [2]Institute of Virology, Philipps University Marburg, Marburg, Germany. [3]Infection Biology Unit, German Primate Center, Göttingen, Germany. [4]Georg-August University Göttingen, Göttingen, Germany. [5]Department of Infectious Diseases, Molecular Virology, University of Heidelberg, Heidelberg, Germany. [6]Biomolecular Medicine, Clinical Research Center, Department of Laboratory Medicine, Karolinska Institutet, Stockholm, Sweden. [7]Department of Paediatrics, University of Oxford, Oxford, UK. [8]Clinic for Gynecology and Obstetrics, Ulm University Medical Center, Ulm, Germany. [9]Department of Medicine, National Jewish Health, Denver, CO, USA. [10]Institute of Virology, Charité-Universitätsmedizin Berlin, Berlin, Germany. [11]Core Facility Extracellular Vesicles, Institute for Tumor Immunology, Center for Tumor Biology and Immunology, Philipps University Marburg, Marburg, Germany. [12]Organisch-Chemisches Institut, Ruprecht-Karls-Universität, Heidelberg, Germany. [13]Department of Tropical Disease Biology, Liverpool School of Tropical Medicine, Liverpool, UK. ✉e-mail: janis.mueller@uni-marburg.de

mechanisms and effectively interact with cellular receptors for entry and replication.

An emerging concept enabling infection and immune evasion of several enveloped (and some naked) viruses is viral apoptotic mimicry[5,6]. Here, viruses benefit from the conserved mechanism where phosphatidylserine (PS) receptors serve as sensors for cells undergoing apoptosis. Apoptotic cells display PS, which is recognized by broadly expressed PS receptors of the T-cell immunoglobulin and mucin (TIM) and Tyro-3, Axl, Mer (TAM) families, resulting in engulfment and anti-inflammatory clearance by surrounding cells[7]. Apoptotic mimicry viruses hijack this immunosuppressive uptake mechanism by exposing PS on the virion, which allows PS-receptor engagement and triggers endocytosis, followed by fusion.

By employing apoptotic mimicry, viruses such as dengue, Zika, Chikungunya, Ebola or Lassa virus may infect a broad spectrum of cell types[5,6]. Despite their broad cell tropism and presence in semen, saliva, human milk, urine and/or blood, transmission via oral or sexual routes is notably limited (Supplementary Table 1). For example, ZIKV RNA and infectious virions have been detected in semen[8–10], breast milk[11] and saliva[10,12,13]. However, sexual transmissions are extremely rare[8,10], breastfeeding transmissions have only been suspected in isolated cases[10,14,15], and there are no reports of oral virus transmission (Supplementary Table 1). This is surprising, considering the presence of infectious virus and abundance of viral RNA in the respective body fluids for extended periods of time and the susceptibility of anogenital and oral cells for infection[16,17].

We recently reported that semen and saliva contain extracellular vesicles (EVs) that competitively block ZIKV binding to target cells and inhibit infection in vitro and ex vivo[16,17]. Reaching up to $10^{10-13}$ particles ml$^{-1}$, EVs are highly abundant in body fluids, especially in semen and saliva[16–19]. However, the underlying mechanism, antiviral spectrum and presence of such EVs in other body fluids remained elusive. There is a substantial overlap in the biogenesis, morphology and cellular attachment mechanism of EVs and viruses[20]. EVs are lipid membrane-enveloped particles released from any cell that carry information in the form of lipids, proteins and nucleic acids. EVs can be transferred or signal to other cells[21] to fulfil various functions[18] including the regulation of immune responses[22]. Notably, PS is enriched in EV membranes[23,24] and, similar to viral apoptotic mimicry, EVs can engage cells via PS-mediated attachment and uptake, without inducing inflammatory responses[20].

In this Article, we show that abundant PS-exposing EVs in body fluids prevent attachment and infection of apoptotic mimicry viruses. Our data reveal a brute-force innate defence mechanism that overpowers viral apoptotic mimicries and may govern viral transmission routes.

## Results

### PS in the viral envelope of ZIKV is essential for infection

Many flaviviruses use viral apoptotic mimicry for cell entry (Supplementary Table 1)[5,6]. Thus, we examined whether PS exposure is also critical for ZIKV infection. To this end, we enzymatically decarboxylated PS headgroups on virions (Fig. 1a), a rapid process reaching a plateau within 5 min (Extended Data Fig. 1a). PS decarboxylation resulted in a significant loss of ZIKV infectivity (Fig. 1b) without cytotoxic effects (Extended Data Fig. 1b). In contrast, HIV-1 and herpes simplex viruses types 1 and 2 (HSV-1/HSV-2), which do not use viral apoptotic mimicry[5,6], maintained infectivity upon enzymatic decarboxylation. Infectivity was even increased by 4.8-fold for HIV-1, 2.1-fold for HSV-1 and 1.6-fold for HSV-2, potentially due to decreased repulsive negative PS charges (Fig. 1b). Hence, PS exposure is crucial for ZIKV infectivity, while it is dispensable for or has a negative impact on HIV-1 and HSV-1/HSV-2.

To further verify ZIKV as viral apoptotic mimicry, we examined whether PS-containing vesicles interfere with infection. We prepared synthetic liposomes containing 0–100 mol% PS in a phosphatidylcholine (PC) background. These liposomes mimic EVs in size

(Extended Data Fig. 2a) and exhibited increasingly negative surface charges (zeta potential) with incorporation of higher amounts of the negatively charged PS (Extended Data Fig. 2b). All liposomes were non-cytotoxic (Extended Data Fig. 2c). While liposomes consisting of only PC had no effect on ZIKV infection, all PS-liposomes displayed dose-dependent inhibition. Their potency increases with the proportion of PS, reaching a plateau with a half-maximal inhibitory concentration (IC$_{50}$) of ~$2 \times 10^9$ particles ml$^{-1}$ for vesicles containing ≥25 mol% PS (Fig. 1c,d). Remarkably, liposomes prepared with only 0.05–0.02 mol% PS displayed inhibitory effects, however, not reaching 50% inhibition at the highest concentrations (~$10^{11}$ particles ml$^{-1}$). We hypothesized that multivalency effects play a role in antiviral efficacy, that is one vesicle may engage multiple PS receptors for improved binding and competition with virions. Indeed, especially at low PS concentrations, antiviral activity was higher in liposomes with unsaturated PC, contributing to membrane fluidity and lateral mobility of PS molecules (Extended Data Fig. 2d). These findings support the notion that ZIKV depends on apoptotic mimicry for efficient infection and reveal that low proportion of vesicular PS is sufficient to interfere with viral infection.

### PS-containing vesicles interfere with virion attachment

We next sought to determine the mechanism by which PS-containing vesicles inhibit ZIKV infection. Time-of-addition experiments revealed that inhibition requires the presence of PS-liposomes before or during viral exposure suggesting an effect on viral entry (Fig. 2a). With increasing amounts of viral inoculum, PS-liposome inhibition potency was reduced (Fig. 2b), indicating that PS-vesicles compete with virions for cell attachment. Lipid headgroups alone phosphoserine, serine or choline did not affect infection (Fig. 2c), suggesting that phosphoserine needs to be exposed on vesicular structures to exhibit antiviral activity. Of note, liposomes exposing phosphatidylethanolamine (PE), a PS-related phospholipid with weak PS-receptor interaction[25], and phosphatidic acid (PA), a different negatively charged phospholipid, exhibited less potent and incomplete inhibitory effects (Fig. 2d). To visualize whether liposomes prevent ZIKV infection by competitively interfering with virion binding, we generated fluorescently labelled PS- and PC-liposomes containing 1 mol% TopFluor-PC. These liposomes were similar in size compared with unlabelled liposomes (Extended Data Fig. 2e) and retained anti-ZIKV activity (Fig. 2e). Confocal microscopy revealed that fluorescent PS- but not PC-liposomes dose-dependently attach to the Vero E6 cell surface and reduce the number of bound Zika virions (Fig. 2f–h). Thus, PS-exposing vesicles inhibit viral apoptotic mimicry infection by competitively interfering with virion cell binding.

### EVs from body fluids expose PS at varying levels

To determine whether endogenous EVs also expose PS and thereby interfere with viral apoptotic mimicry infection, we isolated and analysed EVs from pooled human semen, saliva, urine, breast milk and blood. Source fluids were pre-clarified, and EVs concentrated by tangential flow filtration (TFF) and purified by bind-elute size-exclusion chromatography (BE-SEC; Fig. 3a and Extended Data Fig. 3a). This approach resulted in preparations containing $10^{11}-10^{13}$ particles ml$^{-1}$ as assessed by nanoparticle tracking analysis (NTA; Supplementary Table 2) with narrow size distributions and median diameters ranging from 116 nm (blood EVs) to 160 nm (saliva EVs, Extended Data Fig. 3b). Purification and enrichment of EVs after TFF/BE-SEC was confirmed by sequential increase of the EV-associated CD9 marker for all and Hsp70 and flotillin-1 for most EV sources (Fig. 3b and Supplementary Information) and simultaneous loss of non-vesicular, abundant matrix-specific proteins (Supplementary Information).

To determine PS content, the lipid composition of purified EVs was quantified by shotgun tandem mass spectrometry (MS/MS) lipidomics (Fig. 3c, Extended Data Fig. 3c and Supplementary Table 3). With the exception of blood, PS was the most abundant lipid class in EVs

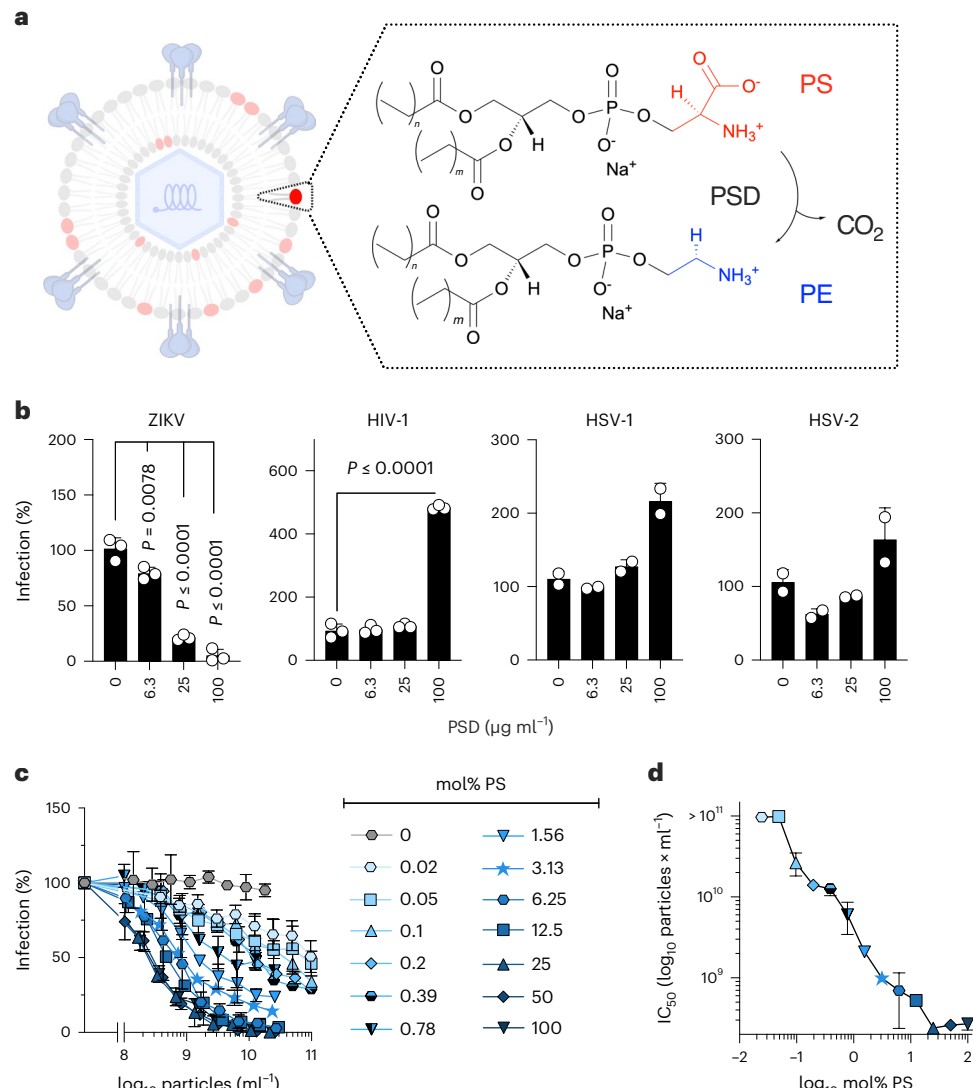

**Fig. 1 | PS exposure in the viral envelope of ZIKV is essential for infection.**
**a**, Schematic enzymatic lipid modification of virions. Viral particles
are treated with PS decarboxylase (PSD), cleaving $CO_2$ and yielding
phosphatidylethanolamine (PE). **b**, Treatment of ZIKV, HIV-1, HSV-1 and HSV-2
with indicated concentrations of PSD for 30 min before inoculation of target
cells. Infection was analysed at 2 days post-infection. $N = 2$ (HSV-1 and HSV-2) or
$n = 3$ (ZIKV and HIV-1) independent experiments with mean biological triplicates,
mean values ± standard deviation (s.d.) shown. One-way ANOVA with Dunnett's
post-test compared with PBS controls, 95% confidence interval, all without
indication $P > 0.05$, not significant. See Extended Data Fig. 1a for reaction kinetics
and Extended Data Fig. 1b for cytotoxicity. **c**, Anti-ZIKV activity of liposomes
containing increasing ratios of PS (the remainder being PC). Cells were exposed
to the indicated concentrations of liposomes and inoculated with ZIKV. $N = 2$
independent experiments with mean biological triplicates, mean values ± s.d.
shown. See Extended Data Fig. 2a–c for generation, characterization and
cytotoxicity of liposomes. **d**, $IC_{50}$ values calculated from data in **c** by non-linear
regression. $n = 2$ independent experiments with mean biological triplicates,
mean values ± s.d. shown.

obtained from all body fluids, reaching up to 57.5 mol% and 56.3 mol%
in semen and urine EVs, respectively. Saliva and breast milk EVs con-
tained intermediate PS concentrations at 31.3 mol% and 17.3 mol%,
while blood EVs contained only 0.1 mol% (Fig. 3c). Remarkably, PS
with 18:0;0/18:1;0 acyl chains was the most abundant lipid molecule
in EVs from semen and urine, constituting ~20% of the entire phos-
pholipid content (Supplementary Table 3). Other major lipid species
included PC (up to 17.7 mol% in blood EVs), sphingomyelin (SM; up to
24.6 mol% in breast milk EVs) and PE (up to 18.8 mol% in breast milk
EVs). While PS was most abundant in semen, PE, which synergistically
enhances PS-receptor binding[25], was prominent in saliva and urine EVs
(Fig. 3d,e). In contrast, blood EVs showed low PS or PE levels but approx-
imately threefold higher PC content compared with saliva or semen
EVs (Fig. 3f). Of note, PS molecules varying in acyl chain length and
saturation exerted the same degree of inhibition when incorporated
into liposomes (Extended Data Fig. 3d). Consequently, all EVs except

those from blood incorporate high ratios of the PS-receptor-binding
lipids PS or PE (Fig. 3g).

Physiologically, PS is usually restricted to the inner leaflet of the
cellular plasma membrane and exposed in a controlled fashion, for
example during apoptosis[26]. Therefore, a fraction of the PS content in
EVs determined by lipidomics may not be exposed and accessible for
competition with virion attachment. We thus used bead-assisted flow
cytometry to capture EVs with typical EV protein markers and stained
for surface-exposed PS using fluorophore-conjugated lactadherin
(LA, Fig. 3h). In each preparation, exposed PS was detected on EVs
bound to anti-CD14 beads. In addition, for all but breast milk EVs, PS was
detected together with major EV markers CD9 and CD63. The variation
of PS-exposure and EV markers between and within fluids underlines EV
heterogeneity and the presence of EV subsets[27]. To quantify the fraction
of PS-exposing EVs in semen, we applied nanoscale flow cytometry and
found that 18.5% of the classical EV marker CD9/CD63/CD81 exposing

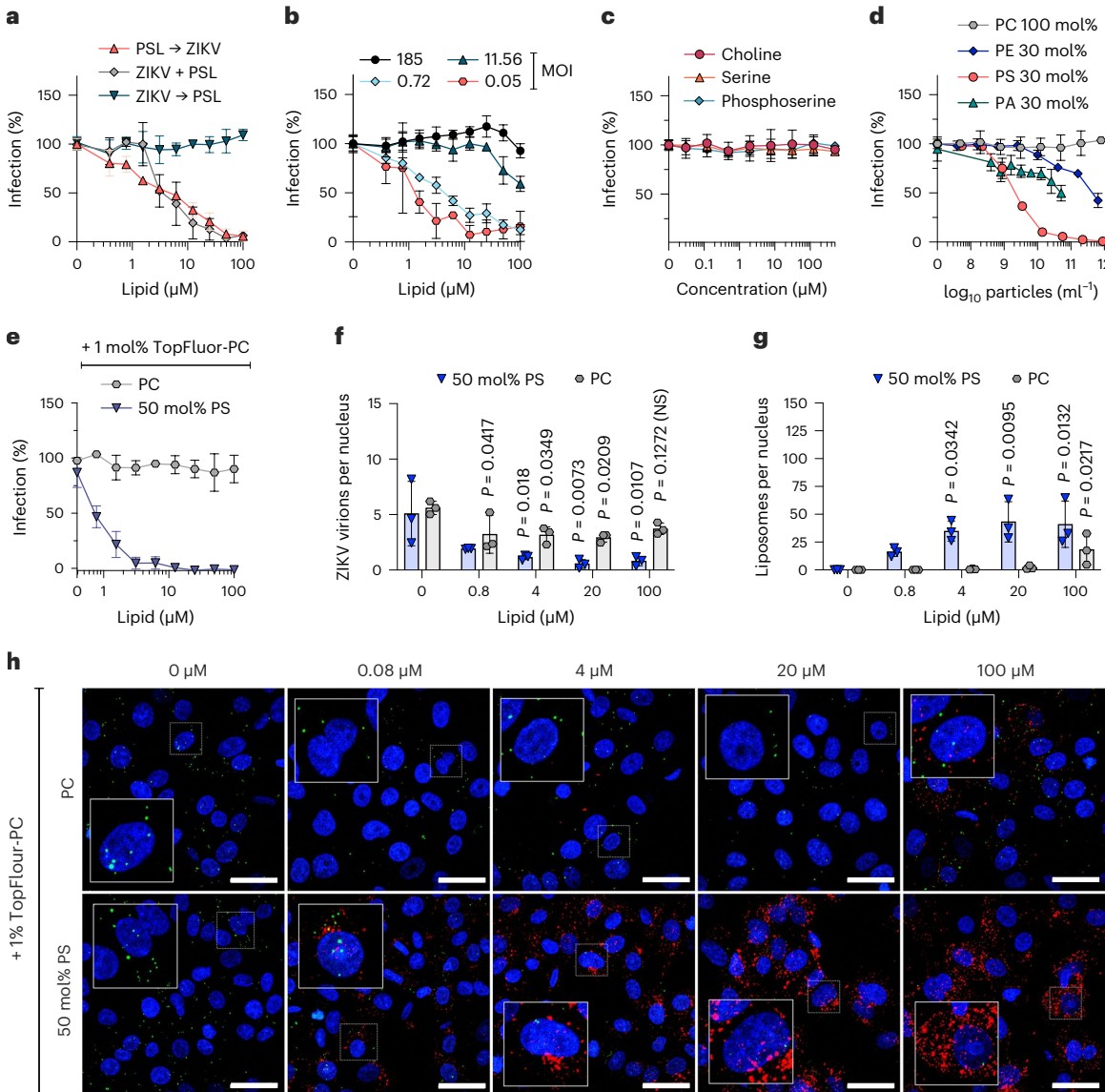

**Fig. 2 | PS-containing vesicles interfere with virion attachment. a**, Time-of-addition virus inhibition experiment with synthetic PS-liposomes (6.25 mol%) added to cells 2 h before, simultaneously with or 2 h after ZIKV. $N = 3$ biological replicates, mean values ± standard deviation (s.d.) shown. **b**, Infection inhibition experiment with titrated synthetic PS-liposomes and a decreasing concentration of viral inoculum indicated as MOI. $N = 3$ biological replicates, mean values ± s.d. shown. **c**, Small molecules equivalent to lipid headgroups tested for antiviral activity against ZIKV. $N = 2$ independent experiments with mean biological triplicates, mean values ± s.d. shown. **d**, Liposomes containing PC only or 30 mol% PE, PA or PS in a PC background tested for antiviral activity against ZIKV. $n = 2$ independent experiments with mean biological triplicates, mean values ± s.d. shown. **e**, Inhibition of ZIKV infection by liposomes containing 99% PC or 50 mol% PS in a PC background with 1 mol% TopFluor. $n = 3$ biological replicates, mean values ± s.d. shown. See Extended Data Fig. 2e for characterization of fluorescent liposomes. **f,g**, Attachment of virions (**f**) or liposomes (**g**) after addition of increasing concentrations of TopFluor-labelled liposomes. Quantification of three separate maximum intensity projections from 25 slices per condition ($n = 3$) performed using Fiji, data shown as means ± s.d. (one-way ANOVA with Bonferroni's post-test, 95% confidence interval). **h**, Representative confocal microscopy maximum intensity projections (of three separate images) of Zika virions (green) and Topfluor (TF)-labelled liposomes (red) on cells (blue nuclei) as used for quantification data shown in **f** and **g**. Scale bars, 20 µm.

EVs displayed LA-detectable PS, indicating a major PS-negative population (Fig. 3i and Extended Data Fig. 3e). Altogether, these findings reveal that EVs isolated from all tested body fluids contain subsets of PS-exposing vesicles, with high levels in semen, saliva and urine and low levels in breast milk and blood.

### PS-rich EVs interfere with viral apoptotic mimicry infection

We next tested the body fluid derived EVs for antiviral activities. All EVs were not cytotoxic (Extended Data Fig. 4a). Semen and saliva EVs inhibited ZIKV infection of Vero E6 cells with $IC_{50}$ values of $6.02 \times 10^{10}$ and $1.7 \times 10^{10}$ particles $ml^{-1}$, respectively (Fig. 4a). This is in line with an incorporation of ≥25 mol% PS at which high antiviral activities were also observed for liposomes (Fig. 1c,d). In comparison, the median $IC_{50}$ for blood EVs of $7.5 \times 10^{11}$ was 12- and 44-fold higher than for semen ($P = 0.0026$) and saliva ($P = 0.002$) EVs (Fig. 4a). Breast milk EVs exhibited an intermediate potency with $IC_{50}$ of $5.8 \times 10^{11}$ particles $ml^{-1}$ (Fig. 4a), respectively. Overall, EVs with high PS exposure showed the highest antiviral activities. Notably, the $IC_{50}$ values of EVs from semen, saliva and breast milk lie well within the physiological particle concentrations in these body fluids (Supplementary Table 2).

In line with receptor binding competition as the mode of action, neither EVs nor PC- or PS-liposomes were directly virucidal

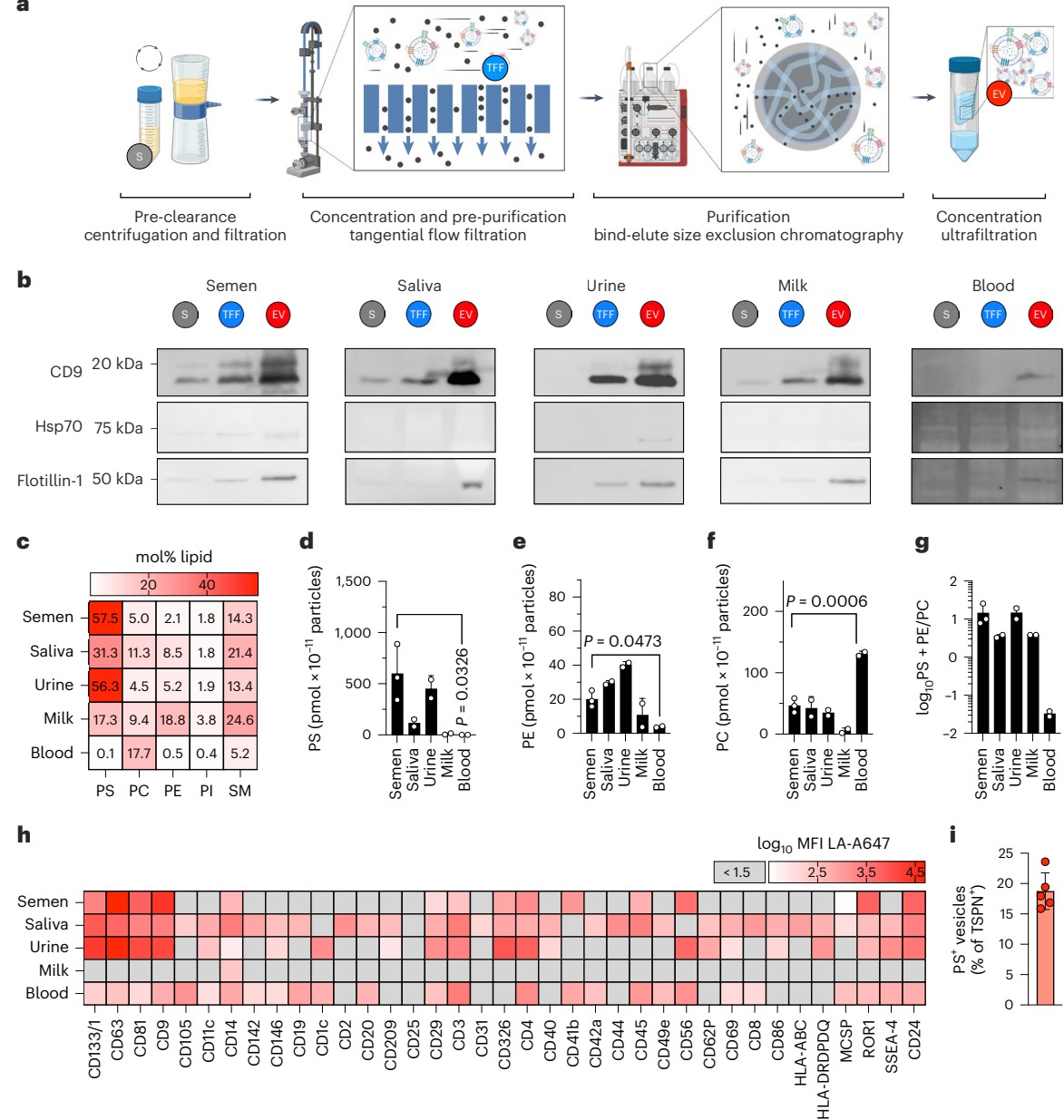

**Fig. 3 | EVs from body fluids expose PS at varying levels. a**, Workflow for purification of EVs by combination of TFF and BE-SEC. See Extended Data Fig. 3a,b for elution profiles and size distributions. **b**, Western blot analysis for verification of EV-associated proteins enrichment by TFF/BE-SEC in protein-concentration-adjusted samples. See Supplementary Information for uncut membranes, markers and total protein staining. Representative blots of $n = 3$ repetitions with similar results for blood, saliva and semen, and $n = 1$ for urine and milk. **c**, Major phospholipid distribution in purified EVs analysed by shotgun lipidomics. Shown are means of two biological replicates with averaged technical duplicates (saliva EVs) or two to three technical replicates of an individual EV preparation (other sources), each from body fluids pooled from at least 35 (semen), 9 (saliva), 6 (urine), an individual (milk) or 50 donors (blood). See Extended Data Fig. 3c and Supplementary Table 3 for extended lipidomics data.

**d**–**f**, Concentration of PS (**d**), PE (**e**) and PC (**f**) according to lipidomics normalized to NTA-based particle concentration per source. One-way ANOVA (95% confidence interval) with Bonferroni's post-test compares semen and blood-derived EVs. **g**, PS + PE to PC ratio based on **d**–**f**. All in **d**–**g** based on lipidomics data from **c** showing means ± s.d. **h**, Surface exposure of PS analysed by bead-assisted flow cytometry for EV subpopulations also exposing the indicated marker proteins. Shown are means of averaged technical duplicates for $n = 5$ (saliva), $n = 3$ (semen, blood) or $n = 1$ (urine, milk) individual EV preparations from pooled sources ± s.d. **i**, Proportion of PS-exposing TSPN-positive (CD9/CD63/CD81 and PS double positive) vesicles in semen EV preparations detected by nanoscale flow cytometry. $n = 5$ biological replicates (individual EV purifications from distinct pooled sources), means ± s.d. (Extended Data Fig. 3e).

(Extended Data Fig. 4b). To visualize the competition of EVs and virions by confocal microscopy, we generated fluorescent hybrid EVs. To this end, labelling-liposomes containing Cy5-conjugated PC were extruded together with semen EVs (Extended Data Fig. 4c). The labelling efficiency was 12.2%, the average vesicle diameter remained unchanged (Extended Data Fig. 4d), and the typical EV marker proteins were detectable by bead-assisted flow cytometry in the Cy5-detecting

channel (Extended Data Fig. 4e). Hybrid semen EVs retained their anti-viral activity (Extended Data Fig. 4f) and attached to cells resulting in dose-dependent inhibition of viral cell binding, while PC-liposomes (Fig. 4b,c) and labelling-liposomes (Extended Data Fig. 4g,h) did not. To assess whether PS-rich EVs interfere with ZIKV attachment by covering viral receptors, we stained cells for Axl, the major PS receptor used by ZIKV[28]. Indeed, signals from hybrid EVs co-localized with stained

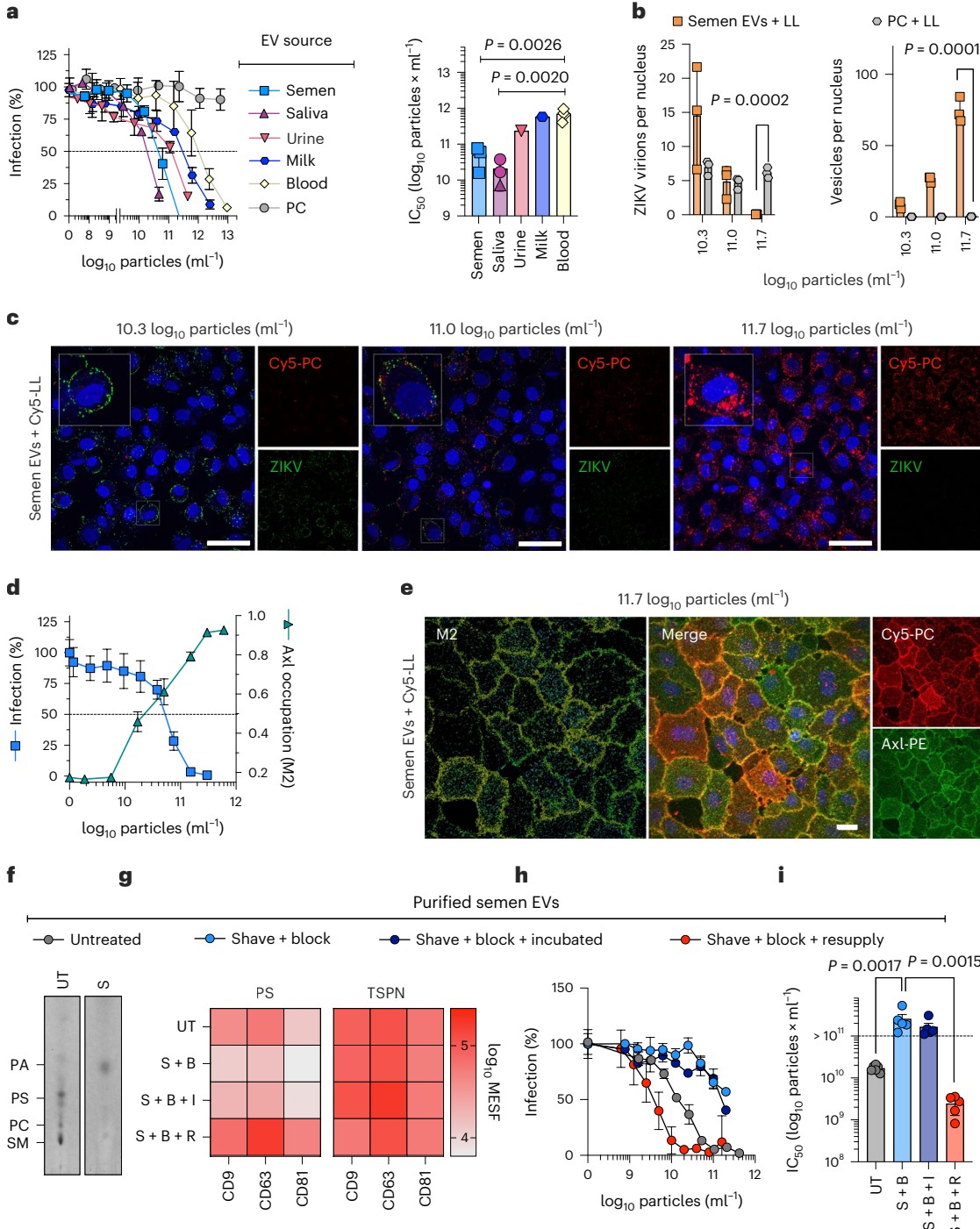

**Fig. 4 | PS-rich EVs from body fluids interfere with viral apoptotic mimicry.**
**a**, Inhibition of ZIKV infection by titrated EVs and control PC-liposomes. $n = 3$
biological replicates, mean values ± standard deviation (s.d.). See Extended
Data Fig. 4a for cytotoxicity. Right: IC$_{50}$ calculated by non-linear regression.
Each data point indicates a separate EV isolation ($n = 4$ (blood), $n = 3$
(semen, saliva), $n = 1$ (urine, milk and liposomes)) analysed in biological
triplicates. One-way ANOVA with Bonferroni's post-test, 95% confidence interval.
**b**, Cell attachment of Zika virions (left) or hybrid vesicles (right) after addition of
Cy5-fluorescent semen EV- or PC-liposome hybrids. See Extended Data Fig. 4c
for labelling workflow and Extended Data Fig. 4d–f for characterization of hybrid
vesicles. Quantification of three maximum-intensity projections from 25 slices
per condition ($n = 3$), mean ± s.d. (for liposomes-only control, see Extended
Data Fig. 4g,h). Unpaired, two-tailed $t$-test, 95% confidence interval. All without
indication NS ($P > 0.05$). **c**, Confocal maximum intensity projections of
Zika virions (green) and hybrid vesicles (red) on cells (blue nuclei) as used
for **b**, representative of three images; scale bars, 20 μm. **d**, Anti-ZIKV (MOI 0.25)

activity of hybrid semen EVs and occupation of Axl determined by confocal
microscopy via M2 overlap coefficients for EV-Cy5 and Axl-PE, means ± standard
error of the mean (s.e.m.) ($n = 3$ z-stacks). Representative image of three shown
in **e**. See Extended Data Fig. 5a for images of all concentrations; scale bar,
20 μm. **f–i**, EVs treated with phospholipase D ('Shave'), LA ('+Block'), followed
by incubation ('+Incubated'). PS reintroduced to EVs by cyclodextrin treatment
('+Resupply'). **f**, Assessment of 'Shaving' (S) by iodine-stained HPTLC.
**g**, Bead-assisted flow cytometry assessing the PS exposure on CD9, CD63 and
CD81 bead-bound EVs by LA staining in comparison with CD9/CD63/CD81
(TSPN) staining. $n = 5$ (Untreated; Shave + Block + Resupply), $n = 4$ (Shave + Block)
or $n = 3$ (Shave + Block + Incubated) separately treated EVs, mean values.
**h**, Anti-ZIKV (MOI 0.25) activity, representative dataset of biological triplicates,
means ± s.d., with corresponding IC$_{50}$ values in **i** for $n = 5$ separately treated
vesicle preparations, means ± standard error of the mean. One-way ANOVA with
Bonferroni's post test (95% CI).

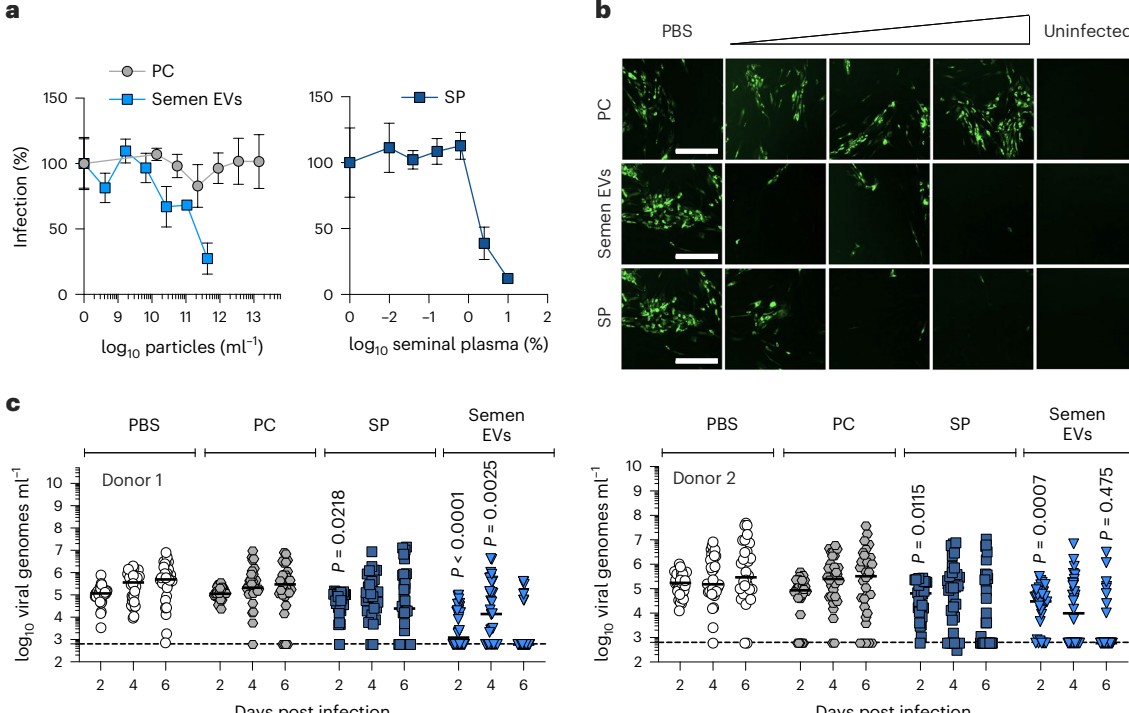

**Fig. 5 | Semen EVs inhibit ZIKV infection of primary cells and tissues from the anogenital tract. a**, Antiviral activity of seminal plasma (SP, right panel), semen EVs and control PC-liposomes (left panel) tested against ZIKV on primary human foreskin fibroblasts. Infection determined by immunofluorescent detection of ZIKV-E and MFI measurements in microwell format (Biotek Cytation). Signals normalized to PBS-treated cells, means of $n = 3$ biological replicates ± standard error of the mean. Representative images (of three replicates) shown in

**b**; scale bars, 500 μm. **c**, Antiviral activity of PC-liposomes ($1 \times 10^{13}$ particles ml$^{-1}$), semen (25 vol%, equivalent to ~0.9 × 10$^{12}$ particles ml$^{-1}$) and semen EVs (4 × 10$^{12}$ particles ml$^{-1}$) tested against ZIKV in ex vivo vaginal tissue of two donors (separated in panels left and right, 32 blocks per condition, line indicates means). Two-way repeated-measures ANOVA with Bonferroni's post-test (95% confidence interval) compares viral genome levels with PBS-treated blocks for the same timepoint; without indication $P > 0.05$, not significant.

Axl on the cell membrane and the signal saturated at the highest concentrations (Mander's M2 coefficient 0.91–0.92), indicating almost complete receptor occupation (Fig. 4d,e and Extended Data Fig. 5a). Notably, with increasing receptor occupation, ZIKV infection correspondingly decreased (Fig. 4d). EVs appeared to be taken up into lysosomes within 1 h of incubation, where they further accumulated over 24 h (Extended Data Fig. 5b,c), suggesting that bound Axl receptors may also be internalized.

To further prove that PS is responsible for the antiviral effect of EVs, we aimed to deplete the exposed PS headgroups from the EV membranes. To this end, we 'shaved' the EVs using the phospholipase D, converting all phospholipids to PA, and 'blocked' remaining PS with LA. Thin-layer chromatography confirmed the conversion and bead-assisted flow cytometry lower PS exposure on all treated samples (Fig. 4f,g). Resulting EVs indeed lost substantial antiviral potency (Fig. 4h,i). This effect was not reversed over time, indicating that remaining PS is, mostly, not shuttled from the inner to the outer membrane layer after treatment. However, resupplying PS to the treated EV membranes using cyclodextrin restored antiviral activity (Fig. 4g–i), confirming PS as the active species in antiviral EVs. Conclusively, EVs in body fluids prevent ZIKV viral apoptotic mimicry infection by displaying PS to interfere with virion attachment but vary in PS content and antiviral potency.

**Physiologic EV concentrations reduce ZIKV infection ex vivo**

To better assess the relevance of EVs in semen as a potential barrier for sexual transmission, we infected ex vivo primary foreskin fibroblasts and vaginal tissue explants in the presence of seminal plasma, semen EVs and PC-liposomes as negative control. ZIKV infection of foreskin fibroblasts was inhibited by seminal plasma with an IC$_{50}$ of

2.9 vol% (corresponding to $1.1 \times 10^{11}$ particles ml$^{-1}$) and by purified EVs with an IC$_{50}$ of $1.5 \times 10^{11}$ particles ml$^{-1}$, while PC-liposomes were inactive (Fig. 5a,b). In vaginal tissues of two donors, productive infection was observed in phosphate-buffered saline (PBS)-treated tissue blocks, with viral RNA levels increasing throughout the course of the experiment for both donors (5.2–5.1 to 5.4–5.7 log$_{10}$ viral genomes ml$^{-1}$, Fig. 5c). Replication did not occur in seminal plasma treated tissues, where RNA titres decreased over time and were only detectable in 22–25% of tissue block supernatants at 6 days post-infection, compared with 94–100% for PBS-treated blocks. Treatment with EVs similarly resulted in absence of viral replication and RNA levels significantly lower than in the PBS control on days 2 and 6 post-infection, indicating that EVs are responsible for the antiviral activity of seminal plasma. Similar to PBS treatment, synthetic PC-liposomes did not inhibit infection (84–88% of tissue blocks infected at day 2) nor replication (increase from 4.9–5.1 to 5.5 log$_{10}$ viral genomes ml$^{-1}$). Overall, physiologic concentrations of semen EVs inhibit ZIKV infection of primary cells and tissues of the genital tract, suggesting that they may act as a barrier to viral transmission at the respective sites.

**EVs inhibit viruses that depend on apoptotic mimicry**

EVs interfering with virus infection via a PS-dependent mechanism should generally compete with viruses exploiting apoptotic mimicry. To challenge this hypothesis, we tested semen EVs, PS-liposomes and control PC-liposomes against a panel of viruses with defined entry strategies. Apoptotic mimicry flaviviruses dengue (DENV) and West Nile viruses (WNV)[5] were inhibited by PS-liposomes and/or semen EVs, but not PC-liposomes (Fig. 6a). Another arthropod-borne apoptotic mimicry, Chikungunya virus (CHIKV) of the *Togaviridae* family[5], was also susceptible to inhibition by EVs and PS-liposomes (Fig. 6a).

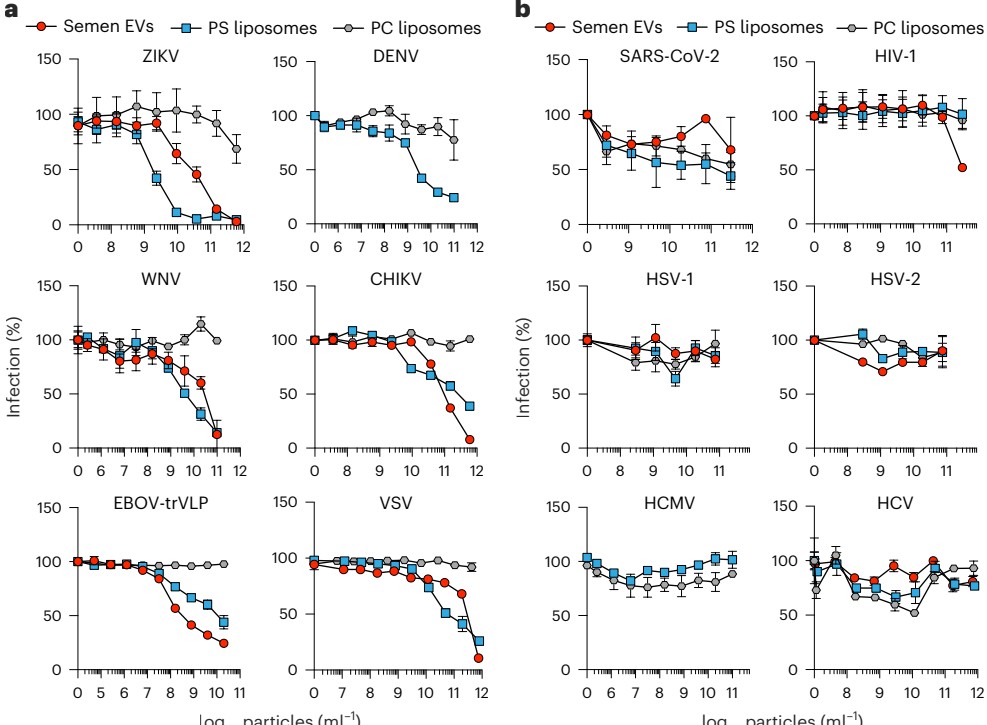

**Fig. 6 | EVs specifically inhibit viral pathogens depending on apoptotic mimicry. a**, Antiviral activity of semen EVs and PS (6.25 mol%)- and PC-liposomes tested against known viral apoptotic mimicry viruses ZIKV, DENV, WNV, CHIKV (all $n = 3$ biological replicates), EBOV(-trVLPs) ($n = 3$ independent experiments with mean biological triplicates each) or VSV ($n = 2$ independent experiments with mean biological triplicates each). **b**, Antiviral activity of semen EVs and

PS- and PC-liposomes tested against viruses with known non-lipid receptors: SARS-CoV-2 ($n = 2$ independent experiments with biological triplicates each), HIV-1 ($n = 3$ biological replicates), HSV-1, HSV-2, HCMV ($n = 2$ independent experiments with biological triplicates each) or HCV ($n = 3$ biological replicates). Shown are mean values ± standard error of the mean.

Similar inhibition was seen for EBOV of the *Filoviridae* family[5] and vesicular stomatitis virus (VSV) of the *Rhabdoviridae* family, both using the strategy of viral apoptotic mimicry[5] (Fig. 6a). This suggests that PS-rich EVs display broad-spectrum activity against viral pathogens using apoptotic mimicry for efficient infection. In contrast, SARS-CoV-2 (ref. 29) and HIV-1 (ref. 30), which do not depend on apoptotic mimicry, were not or only moderately inhibited by PS-vesicles (Fig. 6b). Herpesviruses primarily attach to cells via heparan sulfate proteoglycans[31] and engage multiple receptors[32]. Accordingly, HSV-1/HSV-2, as well as human cytomegalovirus (HCMV), were not inhibited by EVs and/or liposomes. Finally, though belonging to the *Flaviviridae* family with widespread viral apoptotic mimicry, the hepatitis C virus (HCV) is described to require scavenger receptor B1, CD81, claudin-1 or occludin for infection[33]. We show that EVs and PS-liposomes do not interfere with HCV infection, suggesting that, for this hepacivirus, PS interaction is not critical. Together, our data demonstrate that the antiviral activity of EVs in body fluids is broad but specific for apoptotic mimicry viruses.

## Discussion

In the present study, we uncovered an intriguing antiviral defence concept. By mimicking a common viral entry mechanism, biophysically similar EVs overpower viral attachment. Specifically, we show that body fluid EVs expose PS and prevent infection by competing with virions for cell attachment. These EVs are particularly abundant and effective in body fluids that are relevant for oral and sexual virus transmission. Thus, potent inhibition by PS-exposing EVs helps to explain the low incidence of direct human-to-human transmission of apoptotic mimicry viruses.

This discovery is substantiated by several lines of evidence. First, the antiviral activity of semen EVs and seminal plasma was confirmed in primary cells and ex vivo target tissues of ZIKV. Second, ZIKV infection

depends on PS-receptor interaction and EVs interfere with virion attachment not by acting on viral particles, but by occupying cellular PS receptors, followed by EV uptake. Third, this inhibition can be attenuated by depleting and blocking PS on the EV surfaces. In line with this PS-dependent mechanism, EVs interfere with various viruses using apoptotic mimicry for entry, while no activity was observed against viral pathogens using other strategies.

Despite high viral loads in semen, saliva and/or milk, incidences of sexual or oral transmission of arthropod-borne[34] viral mimicries ZIKV[8,10], dengue[11,13], West Nile[11,13] or Chikungunya virus[13,35,36] are low. In contrast, non-viral apoptotic mimicry viruses such as HIV-1, HCMV, HSV-1/HSV-2 or HCV are regularly transmitted via sexual intercourse, or breastfeeding, and SARS-CoV-2 via salivary droplets (Supplementary Table 1). Especially body fluids regularly exchanged between humans contain numbers of PS-exposing EVs exceeding viral particles by several orders of magnitude[16-18]. At the same time, blood-derived EVs show remarkably low amounts of PS exposure and are consequently poorly active against viral apoptotic mimicry infection. Low PS concentrations are a result of the short circulation times of PS-EVs in blood where they are rapidly cleared by phagocytic uptake[37]. Thus, low levels of PS-EVs in blood may contribute to successful blood-borne transmission of viral mimicries either via ingestion by vector species[34] or directly via blood exposure. In line, EBOV efficiently replicates and spreads via blood[38] but is rarely transmitted sexually[39-41]. Interestingly, a recent study describes that also red blood cell-derived EVs interfere with apoptotic mimicry[42], posing the question of whether transient EV shedding might provide sufficient EV concentrations for antiviral competition. Overall, this suggests that PS-rich EVs in body fluids constitute an effective barrier against apoptotic mimicry viruses. This effect might expand to other enveloped viral apoptotic mimicries, such as Lassa virus[5], non-enveloped viruses using vesicular coats such as hepatitis

A virus[43], or even non-viral pathogens exposing PS for invasion, such as plasmodia[44] or *Leishmania*[45].

Our molecular analysis revealed interesting aspects about EV biology and heterogeneity. Liposomes containing only 0.02 mol% PS interfere with viral infection and a fluid membrane background seems to increase antiviral activity by allowing multivalent receptor interactions. However, the saturation or length of PS acyl chains does not affect antiviral activity. Other membrane components might contribute to the antiviral activity of a single EV, such as PE, which synergistically enhances PS-receptor binding[25]. The antiviral potency saturates at PS levels of 25 mol%, a concentration that is still lower than what was detected in semen and saliva EVs. However, in semen 18.5% EVs display PS, highlighting the presence of EV subtypes differing in their antiviral activity. EVs occupy the PS receptor Axl resulting in EV uptake and subsequent receptor internalization may have additional antiviral effects. It will be interesting to decipher what determines uptake and how long cells are protected from infection.

The shared lipid composition and competition for cellular receptors highlights the substantial overlap in biogenesis and cellular attachment mechanisms of EVs and enveloped viruses[20]. Similar to lipids, EVs incorporate proteins shared with viral particles and might compete for other ubiquitous cell surface receptors such as integrins[46] or lectins[47]. This offers room for speculation on the evolutionary origin of EVs and viruses[48]: have viruses emerged from EVs that gained the ability to replicate or are EVs remnants of ancient viruses that have lost it? Similar to viral apoptotic mimicry, EVs utilize the ubiquitous PS-mediated immunosuppressive cell attachment and uptake mechanism[20] and PS exposure in membranes itself drives EV formation[49–51]. Besides apoptotic uptake, involvement of PS in development, cell fusion, coagulation, fertilization and cell–cell communication is evolutionary well conserved[5,26,52]. Consequently, PS receptors are ubiquitously expressed[6,53–56], which expands the viral cell tropism of viral mimicries, such as dengue, Ebola or Lassa virus. Why such a successful entry mechanism and broad tropism is essential for some viruses but dispensable for others is unclear. PS dependence probably involves many factors, such as the subcellular localization of membrane acquisition during biogenesis, or the final fusion mechanism that may occur extracellularly and receptor mediated (HIV-1) or endosomal and pH dependent (ZIKV). PS dependence might also be challenged by the secretion of a high concentration of PS-rich EVs. This might be an evolutionary adaptation of the host against mimicries exploiting the fundamental PS mechanisms that might not be targeted by other means without disrupting the physiologically conserved and diverse functions of PS and PS receptors.

The high levels of PS-exposing EVs in secreted fluids thereby provide an intrinsic antiviral barrier, which is constitutively present and acts even before other innate defence mechanisms, such as the interferon system, are activated. It remains to be fully clarified whether protection against invading pathogens is the main function of PS-EVs or if they also exert important physiological activities. It has been reported that salivary EVs show coagulative properties associated with wound healing[18], but little is known for other body fluids. In pathophysiology, cancer cells secrete PS-rich EVs that modulate the tumour niche, indicating that apoptotic mimicry is involved also in non-infectious diseases[57].

The diverse and fundamental physiological functions of PS make its exploitation for viral entry evolutionarily attractive. At the same time, with overpowering concentrations of PS-containing EVs, the host has developed a counterstrategy that might be translated into clinical interventions. Indeed, blocking viral apoptotic mimicry by an abundance of PS-EVs or synthetic liposomes offers opportunities for broadly effective antiviral strategies. Systemic application or induction of endogenous EV secretion might limit viral apoptotic mimicries that replicate and spread via blood and are highly pathogenic or prevalent such as EBOV showing case fatality rates of ~50% (ref. 58)

or dengue virus infecting 100–400 million individuals each year[59]. Indeed, PS-containing vesicles are already applied in cancer therapy. The liposomal drug Mepact contains 30 mol% PS and is administered systemically without severe adverse effects[60]. Such liposomes or engineered EVs might additionally be optimized by incorporating antiviral or immune stimulating cargo. Targeting apoptotic mimicry not only has the potential to broadly act on viruses but also parasites, bacteria or spread of cancer cells, all of which use exposed PS to persist and spread[44,45,57].

Our study reveals a so far underappreciated innate defence mechanism where PS-exposing EVs effectively compete with and inhibit viral apoptotic mimicries from attaching to cells. This mechanism could explain the selective transmission routes for numerous viruses. Harnessing the antiviral function of PS-EV holds promise for creating a novel class of antivirals, offering broad-spectrum protection—a vital asset against current and emerging viral threats.

We demonstrate that PS exposure is essential for inhibition of apoptotic mimicry viruses and that PS-rich EVs from semen and saliva exhibit greater activity compared with low-PS EVs from blood. Although a strong direct correlation exists between the levels of PS exposure by liposomes and antiviral activity, this correlation does not extend to EVs. One plausible reason is EV heterogeneity as exemplified by 18.5% of semen EVs staining for PS. Further investigation is needed to understand the contribution of EV subtypes with varying PS levels. We show that PS-EVs inhibit ZIKV infection in ex vivo human vaginal tissues, and additional studies in appropriate animal models are required to validate the role of semen EVs in sexual virus transmission. The existing application of PS-liposomes in human therapy indicates tolerability; however, the antiviral benefit warrants further studies.

## Methods

### Ethics statement

Our research complies with all relevant ethical regulations, and all procedures were approved by the ethics committee of Ulm University (88/17, 89/16, 89/17, 337/18 and 131/16). Body fluids were donated by healthy individuals after informed consent by the respective donors. Vaginal tissue was obtained from the Clinic for Gynecology and Obstetrics, Ulm University Medical Center, from surgical removal in patients with pelvic organ prolapse after informed patient consent. Pooled human male AB blood group serum was obtained from Sigma Aldrich (cat. no. H4522).

### Cells

Human embryonic kidney cells (HEK293T; ATCC #CRL3216), HeLa cells expressing CCR5, CXCR4 and HIV-1-tat-responsive β-galactosidase (TZM-bl; NIBSC #ARP5011), human hepatocellular carcinoma cells (Huh-7; provided by R. Bartenschlager, University of Heidelberg), primary human foreskin fibroblasts (HFFs; kindly provided by J. von Einem; Ulm University Medical Center), baby hamster kidney (BHK-21; ATCC #CCL-10) cells, BHK cells expressing HSV-responsive β-galactosidase (ELVIS[61]), and human lung epithelial cells (Calu-3; ATCC #HTB-55; kindly provided by M. Frick, Ulm University) were cultured in Dulbecco's modified Eagle medium (DMEM) supplemented with 10% (v/v) foetal bovine serum (FBS), 100 U ml$^{-1}$ penicillin–streptomycin (Pen-Strep) and 2 mM L-glutamine. African green monkey kidney epithelial cells (Vero E6; ATCC #CRL-1586) were cultured in DMEM supplemented with 2.5% (v/v) FBS, 100 U ml$^{-1}$ Pen-Strep, 2 mM L-glutamine, 1 mM sodium pyruvate and 1% (v/v) non-essential amino acids. All commercially obtained cells were authenticated by the vendor (ATCC or National Institutes of Health AIDS Reagent Program) and not further validated by our laboratory. Cell lines obtained from other groups, who published these cells, were not further authenticated. Ex vivo vaginal tissue was obtained as excess material during surgical procedures and cut into 1 × 1 × 2 cm$^3$ tissue blocks, which were each maintained in 200 μl RPMI1640 medium containing 15% (v/v) FBS, 100 U ml$^{-1}$ Pen-Strep, 2 mM L-glutamine, 1 mM

sodium pyruvate, 100 µg ml⁻¹ gentamicin and 25 µg ml⁻¹ amphotericin B. All cells and ex vivo vaginal tissue blocks were incubated in humidified (95%) incubators at 37 °C and 5% CO₂.

## Viruses

To propagate ZIKV, HSV-1, HSV-2 and SARS-CoV-2, subconfluent Vero E6 cells were inoculated with virus in 5 ml culture medium (for ZIKV, HSV-1 and HSV-2) or 3.5 ml serum-free medium containing 1 µg ml⁻¹ trypsin (for SARS-CoV-2) and incubated for 2 h. Medium containing 10–25 mM HEPES was then added to 40 ml total volume and cytopathic effects were monitored for 3–5 days. To generate HIV-1 virus stocks, subconfluent HEK293T cells were transfected with plasmids encoding full-length viruses using Transit LT-1 (Mirus) at a 3:1 ratio of reagent to DNA. Cells were incubated for 2 days before supernatants were collected. Virus-containing supernatants were generally centrifuged for 3 min at 325g to remove cellular debris, aliquoted and stored at −80 °C as virus stocks. Infectious virus titre was determined by endpoint titration. DENV R2A[16], WNV[16] and transcription- and replication-competent Ebola virus-like particles (EBOV-trVLP)[62,63] were generated as described previously. HCV was produced as described before[64,65], and viral titres were determined by infecting Huh7.5 cells stably expressing the HCV RFP-NLS-IPS reporter[65–67]. VSV was amplified as described before[68,69]. Enhanced green fluorescent protein (EGFP)-expressing CHIKV[70] was produced by in vitro-transcription of and subsequent electroporation of RNA into BHK-21 cells. Virus-containing supernatant was collected, passaged once on BHK-21 cells and viral titres were determined by titration on HEK293T cells.

## EV purification

To purify EVs from non-vesicular proteins and impurities in body fluids, TFF with subsequent BE-SEC was applied as described previously[17,71]. This methodology may be classified as 'intermediate recovery, intermediate specificity' according to MISEV2018 (ref. 72). Source fluids (Supplementary Table 2) were subjected to pre-processing as described in the following. As a standard protocol, source fluids were pre-clarified by low-speed centrifugation in 50 ml polypropylene tubes (Sarstedt) at 700g for 5 min, followed by 3,000g for 5–15 min to remove larger particles and debris.

For purification of EVs from breast milk, a special pre-clearance protocol was followed to sequentially remove the fat-containing fraction, as previously described[73]. Fresh, non-cooled/non-frozen whole human breast milk (stored in Breastmilk Storage Bags, Lansinoh, <4 h after collection at room temperature) was first centrifuged at 3,000g for 10 min at room temperature, and the cream layer was removed. The residual fluid was again centrifuged first at 5,000g (30 min 4 °C), then at 10,000g (30 min 4 °C), each time removing cream after centrifugation, before applying 0.22 µm vacuum filtration and proceeding with EV isolation. For semen, spermatocytes were pelleted by centrifuging at 21,000g for 15–30 min to obtain seminal plasma before proceeding with purification. For saliva, urine and blood, the standard centrifugal protocol was applied.

After filtration, fluids were diluted fourfold in sterile PBS and filtered through a 0.22-µm syringe filter (Millipore Millex-GP, PES) or vacuum filtration units (Millipore SteriCup, PES). The filtrate was diafiltrated with two volumes of sterile PBS and concentrated to ~20–40 ml using a KR2i TFF system (SpectrumLabs) or an Äkta Flux S (Cytiva) with 300 kDa cut-off hollow fibres at a flow rate of 100 ml min⁻¹ (Repligen, transmembrane pressure 3.0 psi; shear rate at 3,700 s⁻¹, mPES). The pre-concentrated sample was subsequently loaded onto two daisy-chained BE-SEC columns (bind-elute size exclusion chromatography; HiScreen Capto Core 700 column, Cytiva) connected to an ÄKTAstart chromatography system (Cytiva) with UNICORN start 1.2.0.164, which was primed (3 ml min⁻¹ flow rate) with 500–1,000 ml sterile PBS beforehand. After loading the entire sample volume, PBS was pumped through the columns (1.5 ml min⁻¹ flow rate) and the EV

sample collected as the first peak in $A_{280}$ absorbance. The collected fraction was then concentrated using a 100 kDa molecular weight cut-off spin-filter (Amicon (RC) or Vivaspin (PES), 3,220g for 60 min). Bovine serum albumin (BSA) was added to 0.1% (v/v) and samples aliquoted to 1.5-ml polypropylene reaction tubes and stored at −80 °C for further downstream analysis. Systems were cleaned with 1 M NaOH in ddH₂O (TFF) or 1 M NaOH in 30% isopropanol (BE-SEC) and rinsed with ddH₂O until the pH of the flowthrough was ~7.

## Nanoparticle tracking analysis

NTA was used to determine concentration, size distribution and surface charge (zeta potential) of particles in body fluids, EV isolations or synthetic liposomes using a ZetaView TWIN (Particle Metrix) as previously described[17]. Samples were diluted in particle-free PBS (for concentration and size distribution), and videos of the scattering particles were recorded with the following settings: 25 °C fixed temperature, 11 positions, 1 cycle, sensitivity 85–90, shutter 100, 15 fps, 2 s videos per position, 3–5 measurements. Videos were evaluated by ZetaView Analyze 08.05.05 SP2 and 08.05.12 SP1. Between the samples, the chamber was thoroughly flushed with particle-free PBS.

## Metabolic activity measurement

Effects of substances or purified EVs on cellular metabolic activity were evaluated using CellTiter-Glo Luminescent Cell Viability Assay (Promega), quantifying intracellular ATP levels. Assays for cell viability determination were conducted in parallel to antiviral assays, with medium only being added instead of virus. CellTiter-Glo assay was performed according to the manufacturer's instructions; medium was removed from the culture, and 50% substrate in PBS was added. After 10 min, luminescence of the samples was measured in an Orion II Microplate Luminometer (Titertek Berthold) using Simplicity 4.2 software. Untreated controls were set to 100% viability.

## Infection assays

**ZIKV and WNV.** ZIKV and WNV infection and its quantification by immunodetection were performed as previously described[16,17]. Briefly, target cells (6,000 Vero E6, A549, HeLa or primary fibroblasts) were seeded in 96-well plates in 100 µl medium per well one day prior and inoculated with virus (multiplicity of infection (MOI) 0.15–0.2) and compounds/vesicles in a total volume of 100 µl. After 2 h, the medium was replaced. Infection was quantified after 2 days by detecting flavivirus E protein with a horseradish peroxidase (HRP)-conjugated secondary antibody. For this, medium was removed and cells were fixed with 4% (para)formaldehyde in PBS (PFA) for 20 min at room temperature. Cells were then permeabilized with 100% ice-cold methanol for 5 min and washed with PBS, before adding 1:10,000 (ZIKV) or 1:5,000 (WNV) diluted mouse anti-flavivirus protein E antibody 4G2 (Absolute Antibody #Ab00230-2.0) in antibody buffer. After 1 h incubation at 37 °C, cells were washed three times with washing buffer before a secondary anti-mouse antibody conjugated to HRP (Thermo Fisher Scientific #31460; 1:10,000) was added and incubated for 1 h at 37 °C. Following four washing steps, TMB peroxidase substrate (Medac) was added and the reaction stopped after 5 min light-protected incubation at room temperature by adding an equal volume of 0.5 M H₂SO₄. The optical density was recorded at 450 nm and baseline corrected for 650 nm using the vMax Kinetic ELISA microplate reader (Molecular Devices) and SoftMax Pro 7.0.3 software. Values were corrected for the background signal derived from uninfected cells, and untreated controls were set to 100% infection. For immunofluorescent imaging of ZIKV infection, cells were fixed and permeabilized in the same manner, but a secondary antibody with fluorescent label was used to detect ZIKV-E (Thermo Fisher Scientific #A11001, diluted 1:400). After washing, nuclei were stained with 1:2,000 diluted Hoechst 33342 (10 µg ml⁻¹ stock in H₂O, Thermo Fisher Scientific #H1399) and incubated for 10 min at room temperature. Imaging was then performed

in a Cytation 3 Microplate Reader/Microscope using Gen5 software (3.04.17, Biotek).

**ZIKV virion attachment assay.** Virion attachment assay was conducted as previously described[16,17]. Vero E6 (150,000 cells per well) were seeded in eight-well μ-slides (Ibidi) 1 day prior, and cells were then inoculated with ZIKV MR766 (MOI 7.4–35) in the presence of putative attachment inhibitors and incubated for 2 h at 4 °C in 300 μl total volume. Cells were then fixed with 4% PFA for 10 min at 4 °C, washed with PBS and unspecific binding sites blocked by 30 min incubation with attachment assay blocking buffer, 5% (v/v) FBS and 1% (v/v) BSA in PBS. Virions were then detected with 1:10,000 diluted mouse anti-flavivirus protein E antibody 4G2 (Absolute Antibody #Ab00230-2.0) in attachment assay binding buffer, PBS with 1% (v/v) BSA for 45 min. After three washing steps with PBS, samples were incubated with 1:1,000 diluted goat anti-mouse secondary antibody conjugated to A647 (Thermo Fisher Scientific #A27029) in binding buffer for 45 min. Finally, cell nuclei were stained with 1:2,000 diluted Hoechst 33342 (Thermo Fisher Scientific #H1399) for 20 min. Attached virus particles were imaged as *z*-stacks of 25 slices of 0.55 μm by confocal microscopy using a Zeiss LSM 710. Images were processed and combined to maximum intensity projections using the ZEN 2.3 (blue edition) software. Attached ZIKV virions were quantified using a custom ImageJ (Fiji Version 2.3.0) macro (Supplementary Code 1) that automatically identifies and counts local fluorescence maxima of a set of 1,024 × 1,024 px confocal images, and co-localization was evaluated using Huygens Professional software (Version 19.10) using Gaussian minimums for thresholding. Pearson's co-localization coefficients are reported as a result, with a value of 0.6 or greater considered a positive correlation.

**ZIKV infection of vaginal tissue blocks.** Vaginal tissues obtained from surgical procedures were cut to $2 \times 2 \times 1 \, mm^3$ tissue blocks, and 32 blocks were incubated with compounds and virus for 2.5 h at 37 °C. After 3× washing with 50 ml PBS, blocks were sorted to individual wells of a 96-well culture plate with 250 μl ex vivo medium[16], 100 μl aas taken as a wash control and replaced by fresh medium. Supernatants were centrifuged at 325*g* for 3 min, transferred to new plates and stored at −80 °C. To study replication, 100 μl supernatant was further collected and replaced by fresh medium on days 2, 4 and 6 post infection and replication finally evaluated by quantitative reverse-transcription polymerase chain reaction as described[17] using StepOne Software 2.3. Primers: RKI-ZIKV-forward 5′-ACGGCYCTYGCTGGAGC-3′; RKI-ZIKV-reverse 5′-GGAATATGACACRCCCTTCAAYCTAAG-3′; ZIKV-probe FAM-AGGCTGAGATGGATGGTGCAAAGGG-BNQ535 (biomers.net)

**DENV.** DENV R2A (Thailand/16681/84)[74,75] infection was performed as previously described[16,74]. Vero E6 cells were seeded 1 day before infection in 96-well plates (6,000 cells per well) and inoculated with virus (MOI 0.15) and compounds/vesicles in a total volume of 200 μl the next day. Infection was determined 3 days post-infection by measuring Renilla luciferase activity.

**EBOV.** EBOV-trVLPs were produced and used in infection assay as previously described[62,63]. Briefly, for EBOV-trVLP production, HEK293T cells (six-well format, 40,000 cells per well) were transfected with expression plasmids for EBOV-NP, EBOV-VP35, EBOV-VP30, EBOV-L, T7-polymerase and the EBOV-trVLP minigenome (p4cis-vRNARLuc). Medium was replaced after 16 h post, and cells were incubated for an additional 48 h, before EBOV-trVLP-containing supernatants were collected and clarified from debris by centrifugation (4,000*g*, 10 min). For infection, HEK293T target cells (96-well plate format, 12,000 cells per well), which were transfected to express EBOV-NP, EBOV-VP30, EBOV-VP35, EBOV-L and DC-SIGN, were seeded 1 day before 96-well plates (12,000 cells per well), inoculated with undiluted EBOV-trVLP supernatant from producer cells and compounds/vesicles in a total volume of 200 μl at

24 h post transfection. Medium was replaced after 6 h and infection was determined at 72 h post inoculation by measuring Renilla luciferase activity in cell lysates. To this end, medium was removed and cells were lysed by incubation with Cell Culture Lysis Reagent (Promega) followed by incubation for 30 min at room temperature. Lysates were then transferred to white 96-well plates, before Renilla luciferase substrate was added (coelenterazine, 2 μM) and luminescence was measured after 60 s incubation. Values were corrected for the background signal derived from uninfected cells, and untreated controls were set to 100% infection.

**SARS-CoV-2.** SARS-CoV-2 infection and its quantification by immunodetection was performed as previously described[17,76]. Calu-3 (15,000 cells per well) target cells were seeded in 96-well plates in 100 μl and the next day replaced with 70 μl of fresh medium. Virus and compounds/vesicles were added in a total volume of 100 μl and 80 μl fresh medium added after 1 h. Infection was quantified after 2 days by detecting SARS-CoV-2 N protein with an HRP-conjugated secondary antibody. Following fixation by adding PFA to a final concentration of 4% and incubating at room temperature for 30 min, medium was discarded and cells permeabilized for 5 min at room temperature by adding 100 μl of 0.1% Triton in PBS. After washing with PBS, 1:5,000 diluted mouse anti-SARS-CoV-2 N protein antibody 40143-MM05 (SinoBiological #40143-MM05) was added in antibody buffer at 37 °C. After 1 h, the cells were washed three times with washing buffer before a secondary anti-mouse antibody conjugated with HRP (Thermo Fisher Scientific #31460) was added (1:15,000) and incubated for 1 h at 37 °C. Following four times of washing, the TMB peroxidase substrate (Medac) was added. After 5 min light-protected incubation at room temperature, reaction was stopped using $0.5 \, M \, H_2SO_4$. The optical density was recorded at 450 nm and baseline corrected for 620 nm using the Asys Expert 96 UV microplate reader (Biochrom). Values were corrected for the background signal derived from uninfected cells, and untreated controls were set to 100% infection.

**HSV-1 and HSV-2.** HSV-1 and HSV-2 infection was determined by measuring green fluorescent protein (GFP) in Vero E6 cells infected with reporter viruses. Vero E6 (6,000 cells per well) were seeded 1 day prior and incubated 1–2 days after addition of virus and test compounds the next day. Medium was then removed and cells were washed once with PBS before adding 0.25% trypsin–ethylenediaminetetraacetic acid for detachment of cells. After incubation at 37 °C and visual confirmation of detachment, an equal volume of medium (containing 10% FBS) was added and cells from wells infected in triplicates were pooled in a 96-well V-bottom plate. After centrifuging 325*g* for 3 min, supernatants were aspirated and cells fixed by adding 4% PFA in PBS and incubating 1 h at room temperature. An equal volume of fluorescence-activated cell sorting buffer was then added, and cells were analysed using a CytoFLEX flow cytometer and CytExpert 2.3 software. Uninfected cells were used as a reference to set gating for the GFP channel, and the percentage of GFP+ cells was used as a quantitative readout for relative infection. Alternatively, infection was determined by ELVIS reporter cells as previously described[77,78].

**HIV-1.** HIV-1 infection was determined using TZM-bl cells, which are HeLa cells engineered to express β-galactosidase under an HIV-responsive promoter. A total of 10,000 TZM-bl cells were seeded 1 day before 96-well plates, virus and test compounds were added the next day, medium was replaced after 2 h and infection was determined at 2 days post-infection. For this, supernatants were removed, Gal-Screen reagent was added and transferred to white well plates following incubation at room temperature, and luminescence was recorded for 0.1 s using a Berthold Microplate reader using Simplicity 4.2 software. Values were background-subtracted (uninfected cells) and normalized to cells in the absence of test compounds.

**HCMV.** HCMV infection was studied on primary HFFs. TB40/E HCMV stocks were kindly provided by Jens von Einem, Institute of Virology, Ulm University Medical Center. HFFs (10,000 cells per well) were seeded 1 day prior, virus and test compounds were added the next day and infection was then determined at 1 day post-infection by immuno-detection of HCMV-IE protein in fixed cells as previously described[79].

**VSV.** VSV–EGFP (kindly provided by KK Conzelmann LMU Munich)[69] infection was performed in Vero E6 cells. A total of 10,000 cells were seeded 1 day before infection in 96-well plates in 100 µl medium per well and inoculated with virus (MOI 0.02) and compounds/vesicles the next day. Medium was replaced after 2 h. One day post-infection, the supernatant was removed, and cells were washed with PBS and detached using trypsin (0.25%) before being transferred to V-bottom plates for fixation (4% PFA, 1 h, 4 °C). Cells were then analysed for GFP fluorescence by flow cytometry (CytoFLEX) using uninfected cells as a control for gating.

**CHIKV.** The CHIKV LR2006-OPY 5′GFP infectious clone express-ing EGFP under the control of a subgenomic promotor (hereafter referred to as CHIKV) has been described previously[70]. A total of 10,000 HEK293T cells were seeded in 96-well plates in 100 µl medium per well and inoculated with CHIKV (MOI 10) and compounds/vesicles on the following day. Viral inoculum was removed and replaced with fresh medium 2 h post infection. One day post-infection, the supernatant was removed, and cells were washed with PBS and detached using trypsin (0.25%) before fixation (2% PFA, 1.5 h, room temperature). Cells were then analysed for the percentage of cells expressing GFP by flow cytometry (FACSLyric) using uninfected cells as a control for gating.

**HCV.** HCV infection rates were analysed by using a Jc1[p7-Gluc-2A-NS2] reporter strain[65,80] and measuring the gaussia luciferase activity in the super-natant. Briefly, 5,000 Huh-7 cells were seeded in 100 µl medium per 96 wells (DMEM (high glucose, sodium pyruvate) supplemented with 10% FBS, 1% glutamine and 1% Pen-Strep) in 96-well plates. The next day, medium was changed to DMEM additionally supplemented with gentamicin (100 µg ml⁻¹) and compounds/vesicles and virus inocu-lum (MOI 0.4) were added. After 2 h, cells were washed once in PBS and further cultivated in fresh DMEM containing gentamicin (200 µl per well). Two days post-infection supernatants were collected and lysed in 1% Triton X-100 (4 °C, 1 h). Secreted *gaussia* luciferase activity was determined using coelenterazine (10 µM, Carl Roth) and a Cen-tro LB 960 luminometer (Berthold Technologies), and mock values were subtracted.

### Virion and EV lipid modifications
**PSD reaction and tracking by DSB-3.** Viruses were treated with PS decarboxylase (referred to as PSD), specifically, a modified, water-soluble and His-tagged version of *Plasmodium knowlesi* also known as His6-Δ34PkPSD for converting surface-exposed PS to PE. The enzyme was produced and purified as previously described[81]. For treatment, PSD was spiked directly into virus stocks at up to 100 µg ml⁻¹ final concentration and reactions incubated at 30 °C while shaking at 450 rpm on an orbital shaker. Reaction kinetics were tracked by a distyrylbenzene-bis-aldehyde (DSB-3) that fluoresces upon reaction with ethanolamine as previously described with slight modification[82]. Briefly, samples of the running PSD reaction were taken at 0 (before addition of enzyme), 5, 15, 30 and 60 min, adding 50 µl sample to a mixture of 37 µl decarboxylation buffer and 12.5 µl of a tetraborate buffer (100 mM sodium tetraborate, pH 9) to stop the enzyme reac-tion, storing the samples at 4 °C until samples from all timepoints were collected. Afterwards, Triton X-100 was added to 1.3 mM final concentration followed by 10 µl of a 100 µM DSB-3 solution. After incu-bation for 60 min at room temperature, fluorescence was measured at $\lambda_{ex}$ = 403 nm and $\lambda_{em}$ = 508 using a microplate fluorescence reader (BioTek Synergy) with Gen5 3.08.01. PSD removal by Ni-NTA bead treatment results in substantial reduced signal in infection inhibition experiments; therefore, PSD-containing samples were diluted tenfold on cells to achieve maximally 10 µg ml⁻¹ on-cell concentrations, which were not cytotoxic (Extended Data Fig. 1b).

**Phospholipase treatment and blockade of surface-exposed PS.** To abrogate the antiviral activity of EVs, surface-exposed PS (and other phospholipids) were first 'shaved' by digestion with phospholipase D (PLD from *Streptomyces chromofuscus*, Sigma Aldrich #P0065). For this, EVs purified using TFF/BE-SEC as described in Tris buffer (50 mM Tris base and 100 mM NaCl, adjusted to pH 7.4 with HCl) were digested with 500 U PLD in the presence of 10 mM CaCl₂ on a tube rotator at 40 °C. PLD was then removed by CaptoCore700 BE-SEC and vesicles concentrated by 100 kDa ultrafiltration as for initial purifica-tion. To block residual surface-exposed PS, vesicles were additionally incubated with bovine LA (Cellsystems, #BLAC-1200) at 50 µg ml⁻¹ overnight. Antiviral activity of 'shaved & blocked' EVs was then tested directly or after incubation for 1 week at 4 °C followed by 1 day at 37 °C ('+ Incubated').

**Cyclodextrin-mediated lipid exchange.** PS-depleted vesi-cles were re-supplied with PS to restore antiviral activity by using cyclodextrin-mediated lipid exchange. Methyl-α-cyclodextrin (Ara-Chem #CdexA-076/BR) was incubated with donor liposomes consist-ing of 100 mol% DOPS at a ratio of 1.5 mM lipid to 40 mM cyclodextrin for 2 h while on a tube rotator at 40 °C. Lipid-loaded cyclodextrins were then separated from the donor liposomes by ultrafiltration with a 10 kDa MWCO (Amicon 0.5, Merck Millipore). Loaded cyclo-dextrins were then mixed with PS-depleted vesicles (350 µl loaded cyclodextrin filtration flowthrough + 50 µl depleted vesicles + 100 µl PBS) and samples incubated for 2 h while on a tube rotator at 40 °C. Cyclodextrins were then again separated by ultrafiltration with 10 kDa MWCO as before, and the retentate containing lipid-exchanged vesicles was eluted.

### Confocal microscopy for hybrid EV receptor occupation and uptake assay
For analysing EV receptor occupation, Vero E6 (70,000 cells per well) were seeded in eight-well µ-slides (Ibidi) 1 day prior. Cy5-labelled hybrid semen EVs were then added at indicated concentrations and slides incubated 2 h at 4 °C. Supernatants were then removed and cells fixed by addition of 4% PFA for 10 min at room temperature followed by addition of anti-Axl-PE (Thermo Fisher Scientific #12-1087-42, diluted in PBS + 1 vol% FBS to 120 ng ml⁻¹) and incubation at 37 °C for 1 h. Cells were then washed 3× with PBS + 1 vol% FBS before staining nuclei with 1:2,000 diluted Hoechst 33342 (Thermo Fisher Scientific #H1399). After two more washing steps, cells were imaged as *z*-stacks of 20 slices of 0.55 µm by confocal microscopy using a Zeiss LSM 710 and ZEN 2.3 (blue edition) software. *z*-Stacks were analysed by Huygens Professional software (Version 19.10) using Gaussian minimums for thresholding. Mander's M2 overlap coefficient (reporting the frac-tion of PE/Axl-positive pixels that are also Cy5/SemenEV positive) is reported as a result.

For hybrid EV uptake assay, 40,000 Vero E6 cells were seeded to a 35 µ-dish (Ibidi) in 2 ml medium 1 day prior. The next day, LysoTracker Green DND-26 (Thermo Fisher, #L7526) was added at 75 nM in fresh medium and cells were incubated for 30 min before replacing the medium with phenol-red free DMEM (supplemented as the normal growth medium), adding hybrid semen EVs at 5 × 10¹⁰ particles ml⁻¹ and imaging the entire cell volume as *z*-stack immediately using a Leica SP8 confocal microscope in resonance mode. Cells were incu-bated between imaging timepoints, and the gain corresponding to LysoTracker Green DND-26 was increased to compensate for fading during long-term imaging.

## Phospholipid HPTLC with iodine vapor staining

To analyse lipid content of liposomes and EVs by thin-layer chromatography, total lipid was isolated using the Folch method. For this, 25 µl sample was mixed with 500 µl solvent consisting of a 2:1 (v/v) ratio of chloroform and methanol, followed by 5 min of sonication in a bath sonicator and 2,000 rpm shaking for 20 min at 4 °C. Then 125 µl water was added to induce phase separation, and samples were centrifuged at 2,000$g$ for 5 min. The upper aqueous layer was removed, and the lower phase was transferred to new vials. Solvent was then evaporated under nitrogen, and 25 µl chloroform was added before shaking for 5 min at 2,000 rpm and sonicating for 5 min. High-performance thin-layer chromatography (HPTLC) plates (Sigma-Aldrich) were activated by heating to 110 °C for 30 min, and samples were then spotted in 3 µl volume using graduated glass pipettes (Hirschmann Ringcaps). The running chamber was filled with mobile phase, consisting of methyl acetate, isopropanol, chloroform, methanol and aqueous KCl (0.25) at a ratio of 25:25:25:10:9, as described previously[83]. Plates were then inserted and allowed to run until the solvent front had reached within 0.5 cm of the upper plate limit. Plates were then air dried for 10 min and developed in an iodine vapour chamber. After 30 min, plates were immediately imaged in a GelDoc (Bio-Rad).

## Shotgun lipidomics

Mass spectrometry (MS)-based lipid analysis was performed by CRO Lipotype GmbH as described[84]. Lipids were extracted using a two-step chloroform/methanol procedure[85]. Samples were spiked with internal lipid standard mixture containing: cardiolipin 14:0/14:0/14:0/14:0 (CL), ceramide 18:1;2/17:0 (Cer), diacylglycerol 17:0/17:0 (DAG), hexosylceramide 18:1;2/12:0 (HexCer), lyso-phosphatidate 17:0 (LPA), lyso-phosphatidylcholine 12:0 (LPC), lyso-phosphatidylethanolamine 17:1 (LPE), lyso-phosphatidylglycerol 17:1 (LPG), lyso-phosphatidylinositol 17:1 (LPI), lyso-phosphatidylserine 17:1 (LPS), phosphatidate 17:0/17:0 (PA), phosphatidylcholine 17:0/17:0 (PC), phosphatidylethanolamine 17:0/17:0 (PE), phosphatidylglycerol 17:0/17:0 (PG), phosphatidylinositol 16:0/16:0 (PI), phosphatidylserine 17:0/17:0 (PS), cholesterol ester 20:0 (CE), sphingomyelin 18:1;2/12:0;0 (SM) and triacylglycerol 17:0/17:0/17:0 (TAG). After extraction, the organic phase was transferred to an infusion plate and dried in a speed vacuum concentrator. First-step dry extract was resuspended in 7.5 mM ammonium acetate in chloroform/methanol/propanol (1:2:4, v:v:v) and second-step dry extract in 33% ethanol solution of methylamine in chloroform/methanol (0.003:5:1; v:v:v). All liquid handling steps were performed using Hamilton Robotics STARlet robotic platform with the Anti Droplet Control feature for organic solvents pipetting. Samples were analysed by direct infusion on a Qexactive mass spectrometer (Thermo Scientific) equipped with a TriVersa NanoMate ion source (Advion Biosciences). Samples were analysed in both positive and negative ion modes with a resolution of $R_{m/z = 200} = 280,000$ for MS and $R_{m/z = 200} = 17,500$ for MS/MS experiments, in a single acquisition. MS/MS was triggered by an inclusion list encompassing corresponding MS mass ranges scanned in 1 Da increments[86]. Both MS and MS/MS data were combined to monitor CE, DAG and TAG ions as ammonium adducts; PC and PC O-, as acetate adducts; and CL, PA, PE, PE O-, PG, PI and PS as deprotonated anions. MS only was used to monitor LPA, LPE, LPE O-, LPI and LPS as deprotonated anions; Cer, HexCer, SM, LPC and LPC O- as acetate adducts. Data presented as "mol% lipid" herein indicate the proportion of specific lipid species amongst all analysed. Cholesterol was not included in our analysis. Data were analysed with in-house developed lipid identification software based on LipidXplorer[87,88]. Data post-processing and normalization were performed using an in-house developed data management system. Only lipid identifications with a signal-to-noise ratio >5 and a signal intensity fivefold higher than in corresponding blank samples were considered for further data analysis.

## Multiplex bead-based EV surface protein profiling by flow cytometry

Purified EVs were subjected to bead-based multiplex EV analysis (MACSPlex Exosome Kit, human, Miltenyi Biotec) as previously described[17,89] with adaptations. Briefly, EVs purified by TFF/BE-SEC were diluted at input doses of $10^9$ or $10^{10}$ NTA-quantified particles per assay in MACSplex buffer and incubated overnight with MACSPlex Exosome Capture Beads on an orbital shaker at 450 rpm at room temperature. Beads were washed with MACSPlex buffer, and most of the supernatant was aspirated. For staining of captured EVs, a cocktail of APC-conjugated anti-CD9, anti-CD63 or anti-CD81 detection antibodies (Tetraspanins (TSPN); Miltenyi Biotec; #130-108-813; 5 µl each) or alternatively for detection of PS, 0.5 µg LA-A647 (Haematologic Technologies, #BLAC-ALEXA647) were added tube followed by incubation at 450 rpm for 1 h at room temperature in the dark. Next, the samples were washed twice with MACSplex buffer and liquid removed before resuspension in 150 µl MACSPlex buffer. Samples were then transferred to a V-bottom 96-well microtitre plate (Thermo Scientific) and analysed by flow cytometry using a Cytoflex LX (Beckman Coulter). CytExpert 2.3 (Beckman Coulter) was used to analyse flow cytometric data (see Supplementary Information for gating strategy). Median fluorescence intensities (MFIs) for all 39 capture bead subsets were background-corrected by subtracting respective MFI values from matched non-EV containing buffer controls that were treated exactly like EV samples (buffer + capture beads + antibodies). Raw MFI values below those obtained for isotype controls of each respective sample are reported as 'not detected'. For experiments directly comparing samples not measured on the same day (Fig. 4g), MFI was converted to molecules of equivalent soluble fluorochrome (MESF) by calibration beads (Bangs Laboratories Quantum #647 for calibrating LA A647 data; #823 for calibrating TSPN-APC data) to ensure consistency.

## Nano-flow cytometry

For nano-flow cytometry, a Flow NanoAnalyzer (NanoFCM) equipped with a 488 nm and a 638 nm laser, was calibrated using 200 nm polystyrene beads (NanoFCM) with a defined concentration of $2.08 \times 10^8$ particles ml⁻¹, which were also used as a reference for particle concentration. In addition, monodisperse silica beads (NanoFCM) of four different diameters (68 nm, 91 nm, 113 nm and 155 nm) served as size reference standards. Freshly filtered (0.1 µm) 1× TE (Tris-EDTA) buffer pH 7.4 (Lonza) was analysed to define the background signal, which was subtracted from all other measurements. Each distribution histogram or dot plot was derived from data collected for 1 min with a sample pressure of 1.0 kPa. The EV samples were diluted with filtered (0.1 µm) 1× TE buffer, resulting in a particle count in the optimal range of 2,500–12,000 events. Particle concentration and size distribution were calculated using NanoFCM software (NF Profession V2.0). For immunofluorescent staining, 12.5 µM of corresponding antibodies in 50 µl TE was used: fluorescein isothiocyanate (FITC)-conjugated mouse anti-human CD9 antibody (clone HI9a; BioLegend #312104), FITC-conjugated mouse anti-human CD81 antibody (clone TAPA-1; BioLegend #349504) and FITC-conjugated mouse anti-human CD63 antibody (clone H5C6; BioLegend #353005); as isotype controls, FITC-conjugated mouse IgG1, κ (clone MOCP-21; BioLegend #400109) and FITC-conjugated mouse IgG2a, κ (MOPC-173; BioLegend #400207), 2 ng µl⁻¹ of each antibody in 50 µl 1× TE buffer. PS-staining was done using 5 µg ml⁻¹ LA-A647 (Haematologic Technologies, #BLAC-ALEXA647). After removing antibody aggregates by centrifugation at 12,000$g$ for 10 min, the supernatant was added to $2 \times 10^8$ particles, followed by incubation for 12 h at 37 °C under constant shaking and washing with 1 ml 1× TE buffer by ultracentrifugation at 110,000$g$ for 45 min at 4 °C (Beckman Coulter MAX-XP centrifuge, TLA-45 rotor; Beckman Coulter). The pellet was resuspended in 50 µl 1× TE buffer for nano-flow cytometric analysis.

## Liposome synthesis

Liposomes were prepared by thin-film hydration and extrusion as described[17,90]. For this, lipids in chloroform were added to a glass round-bottom flask and the solvent was then evaporated by slowly applying a vacuum at a Schlenk line. The vacuum was held for 2 h and then purged with argon. Alternatively, solvent was evaporated under a nitrogen gas stream for at least 1 h at room temperature and the lipid film then overlaid with argon. Next, the lipid film was hydrated by adding PBS and the flasks were shaken at 180 rpm at a temperature dependent on the transition temperature of the lipids utilized. Small unilamellar vesicles were then prepared by at least 25× extrusion through 0.2 µm polycarbonate membranes (Nuclepore Track-Etched Membrane, Whatman) in a Mini Extruder (Avanti Polar Lipids); for lipid compositions with transition temperature above room temperature this was done on a heating block set to the respective temperature. Liposomes were quantified by NTA using a ZetaView (ParticleMetrix).

## Hybrid EV-liposome preparation

To label TFF/BE-SEC-purified EVs with a synthetic dye-conjugated lipid, a lipid mixture containing 90 mol% dioleoyl PC and 10 mol% Cy5-dioleoylPC was added to glass vials and the solvent was evaporated as described above. To 100 µM dried lipid mixture, $2.5 \times 10^{12}$ EVs (or unlabelled liposomes) were then added and the mixtures were shaken at 45 °C, 180 rpm for 1 h. Serial extrusion was then done 5× through 1 µm, 10× through 0.2 µm and 20× through 0.1 µm membranes. Vesicles were then stored at −80 °C under argon until use.

## Determination of protein content by BCA assay

The protein content of samples was quantified using Pierce Rapid Gold BCA Protein Assay Kit as described by the manufacturer (Thermo Fisher) using the microwell procedure using using the vMax Kinetic ELISA microplate reader (Molecular Devices) and SoftMax Pro 7.0.3 software.

## SDS-PAGE

Sodium dodecyl sulfate polyacrylamide gel electrophoresis and total protein staining were performed as previously described[17]. Protein samples were mixed with protein loading buffer (LI-COR) and tris(2-carboxyethyl)phosphine as reducing agent (50 mM final concentration) and heated to 70 °C for 10 min. Proteins were then separated on NuPAGE 4–12% BisTris gels (200 V, 30 min). For total protein staining, gels were fixed with a 50% methanol:7% acetic acid solution and stained with GelCode Blue (Thermo Fisher) for 1 h at room temperature. After destaining with ultrapure water, the gel was imaged on a LI-COR Odyssey near-infrared imager or a ChemiDoc Imaging Systems.

## Western blot

Western blot analysis of EV and non-EV marker proteins was performed as previously described[17]. Briefly, protein concentrations of samples were first measured by bicinchoninic acid assay (BCA) as described above and then adjusted to that of the lowest-concentrated sample with PBS. Concentration-adjusted samples were mixed with protein loading buffer (LI-COR) and tris(2-carboxyethyl)phosphine (Sigma Aldrich, 50 mM final concentration) and heated to 70 °C for 10 min. Proteins were then separated on NuPAGE 4–12% BisTris gels, blotted onto Immobilon-FL polyvinylidene fluoride membranes via semi-dry transfer and blocked with 0.25% casein in PBS-T 0.05% (Thermo Scientific). Membranes were then stained with primary antibodies (CD9, 1:1,000, Cell Signaling Technology #13174; flotillin-1, 1:1,000, Cell Signaling Technology, #18634; HSP70, 1:1,000, Cell Signaling Technology, #4876) in 0.025% casein in PBS-T 0.05% (staining buffer) overnight a 4 °C. The next day, membranes were washed 3× with PBS-T 0.05% and incubated with secondary antibody (1:2,500, Bio-Rad #12005870; #12004162 or 1:10,000 Thermo Fisher Scientific #31460) in staining buffer for 1 h at room temperature. After 4× washing, membranes were

imaged using a ChemiDoc Imaging Systems measuring fluorescence or chemiluminescence (ECL Substrate, Bio-Rad #1705062).

## Statistics

No data points were excluded from the analyses. Data collection was not randomized, and collection and analysis were not performed blind to the conditions of the experiments. No statistical methods were used to pre-determine sample sizes, but our sample sizes are similar to those reported in previous publications[16,17]. Data distribution was assumed to be normal, but this was not formally tested. Data were analysed using Microsoft Excel for Mac, Version 16.71 and analysed and visualized using GraphPad Prism for macOS Version 9.5.1. Statistical analyses were also performed using GraphPad Prism for macOS Version 9.5.1 and 10.0.2. $IC_{50}$ concentrations were calculated by non-linear regression ([Inhibitor] versus normalized response). Differences between treatment groups were tested for statistical significance by one-way analysis of variance (ANOVA) with Bonferroni's post-test or two-way repeated-measures ANOVA with Dunnett's post-test when multiple timepoints were included in the analysis, unless stated otherwise. Details on statistical details of individual experiments are included in the figure legends.

Illustrations in Figs. 1a and 3a and Extended Data Fig. 4c were created with BioRender.com.

### Reporting summary

Further information on research design is available in the Nature Portfolio Reporting Summary linked to this article.

## Data availability

The data generated during this study are provided as Source data and Supplementary Code. Any additional information required is available from the lead contact upon request. No unique/stable reagents and materials were generated in this study. Source data are provided with this paper.

## Code availability

The custom ImageJ/Fiji macro is attached (Supplementary Code 1).

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

## Acknowledgements

We thank all body fluid and tissue donors. We thank M. Mayer, D. Krnavek, N. Schrott, R. Burger, J. Romana Fischer and B. Ott for skilful laboratory assistance. We are thankful to T. Hoenen (Friedrich-Loeffler-Institut, Greifswald, Germany) for sharing the EBOV-trVLP system. Thanks to J. Kumpf and K. Brödner (Organisch-Chemisches Institut, University of Heidelberg) for the synthesis and purification of DSB-3. Thanks to J. von Einem (Institute of Virology, Ulm University Medical Center), M. Frick (Institute of General Physiology, Ulm University) and K.-K. Conzelmann (Max von Pettenkofer Institute Virology, LMU Munich) for providing cells and viruses. J.A.M. receives funding for this project by the German Research Foundation (MU 4485/1-1). K.M.J.S. is funded by the BMBF (IMMUNOMOD-01KI2014), F.K. by an Advanced ERC grant (Traitor-viruses), and J.M. by the Carl-Zeiss Stiftung (Ultrasensvir). J.M., F.K. and K.M.J.S. receive funding by the German Research Foundation via Project ID 316249678(SFB 1279) and K.M.J.S. additionally via SP 1600/7-1 and 1600/9-1. J.M. further acknowledges funding by MWK Baden-Württemberg (MWK33-7532-56/12/32). R.G. acknowledges funding by the Bausteinprogramm of Ulm University Medical Faculty (L.SBN.0223). S.P. acknowledges funding by BMBF (01KI2006D, 01KI20328A and 01KX2021), the Ministry for Science and Culture of Lower Saxony (14-76103-184, COFONI Network, including projects 7FF22, 6FF22 and 10FF22), EU (project UNDINE) and the German Research Foundation (DFG; PO 716/11-1 and PO 716/14-1). C.G. receives funding by the German Research Foundation (GO2153/6-1 and GO2153/8-1).

## Author contributions

Conceptualization: R.G., J.M. and J.A.M.; investigation: R.G., H.R., P.v.M., D.A., L.S., H.B., M.H., M.C., J.J., C.P. and J.A.M.; resources: D.G., M.D., J.-Y.C., K.S., S.P., D.R.V., C.G., E.P.-v.s., U.B., R.B., S.E.A., K.M.J.S., E.H., S.B., F.K., J.M. and J.A.M.; data curation: R.G. and J.A.M.;

writing—original draft: R.G. and J.A.M.; writing—review and editing: all authors; visualization: R.G. and J.A.M.; supervision: R.G., J.M. and J.A.M.; funding acquisition: J.M., K.M.J.S., F.K., R.G. and J.A.M.

## Competing interests

The authors declare no competing interests.

## Additional information

**Extended data** is available for this paper at https://doi.org/10.1038/s41564-024-01637-6.

**Correspondence and requests for materials** should be addressed to Janis A. Müller.

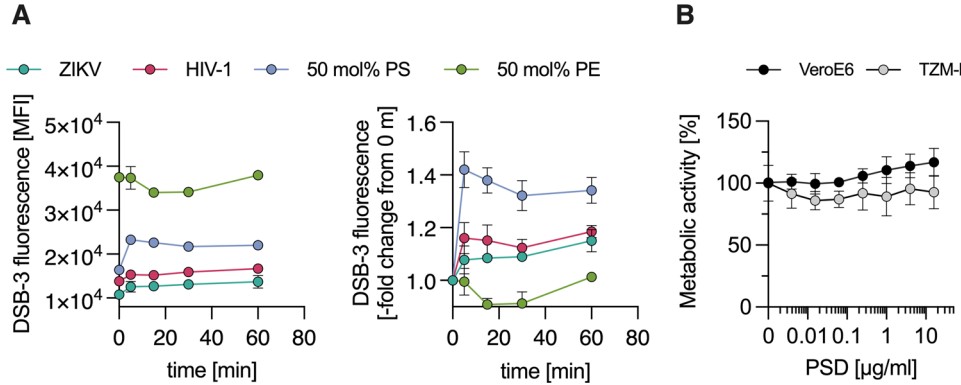

**Extended Data Fig. 1 | PS decarboxylase reaction kinetics and cytotoxicity control.** (a) Kinetics of PS decarboxylase (PSD) PS to PE conversion on viruses determined using DSB-3, an amine-binding fluorophore. Virus stocks were incubated with PSD (50 µg/ml) for up to 60 min at 30 °C while shaking and samples taken at indicated time intervals. At each sampling time point, an aliquot was taken and the reaction stopped by adding sodium tetraborate buffer (pH 9) and storage at 4 °C until DSB-3 staining. Samples were lysed using 1.5 mM triton X-100, stained with 10 µM DSB-3 for 1 h and fluorescence quantified in a microplate reader. 50 mol% PS and PE liposomes serve as controls. Raw data shown in the left panel, data normalized to the first time point (0 min, that is before addition of PSD) shown in the right panel. n = 2 (HIV-1, liposomes) or n = 3 independent reactions (ZIKV), means ± SD. (b) Metabolic activity of Vero E6 (used for ZIKV, HSV-1, HSV-2) and TZM-bl cells (used for HIV-1) after treatment with PSD concentrations equivalent to those used in antiviral experiments in Fig. 1b. n = 2 independent experiments in biological triplicates each, means ± SD.

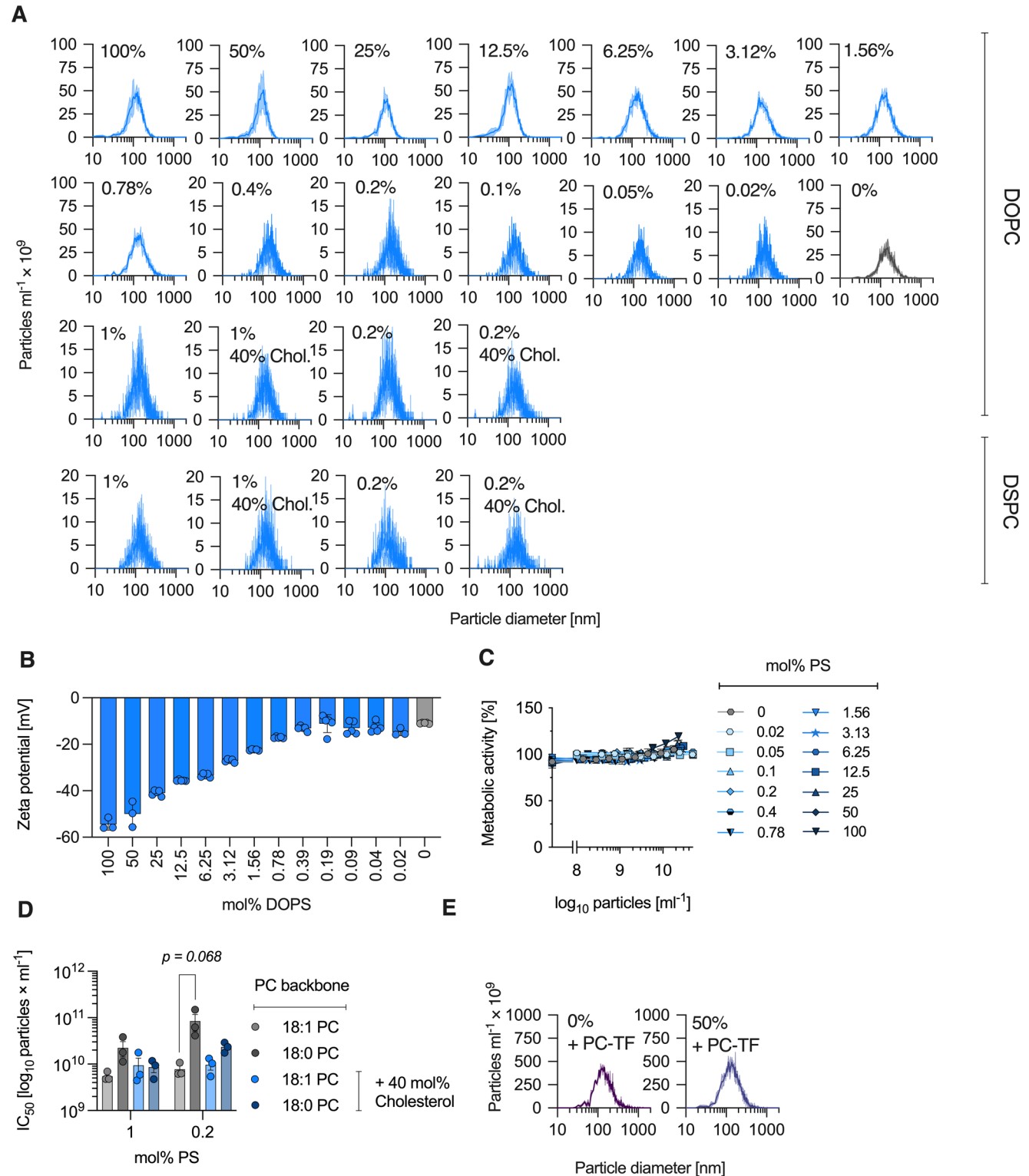

**Extended Data Fig. 2 | Liposome characterization. (a)** Representative NTA histograms showing size distribution detected for liposome preparations with indicated molar percentage of PS in dipalmitoyl-PC (DOPC) or distearoyl-PC (DSPC) backbones. Lines show mean and shadings SD of 3 replicate acquisitions. **(b)** Surface charge of liposomes measured as Zeta-potential in 1 mM KCl. n = 5 or n = 3 (for 100, 50 and 0.02%) measurements per preparation, data show means ± SD. **(c)** Metabolic activity of Vero E6 cells incubated with liposomes containing different molar ratios of PS in a PC backbone as used in antiviral experiments in Fig. 1c. n = 3 biological replicates, means ± SD. **(d)** Half-maximal inhibitory concentrations (IC$_{50}$) (ZIKV-MR766 MOI 0.25 on Vero E6) of liposomes containing 1 or 0.2 mol% PS in a 18:0 (DSPC) or 18:1 (DOPC)-PC backbone, with or without addition of 40 mol% cholesterol. n = 3 biological replicates, data show means ± SD. IC$_{50}$ values compared using ordinary one-way ANOVA with Bonferroni's post test (95% CI); all comparisons p > 0.05, those without specific indication > 0.01. **(e)** NTA histograms of fluorescent (1% PC-TopFluor (PC-TF) containing) liposomes containing 0 or 50 ml% PS used in attachment assays. Lines show mean and shadings SD of n = 3 replicate acquisitions.

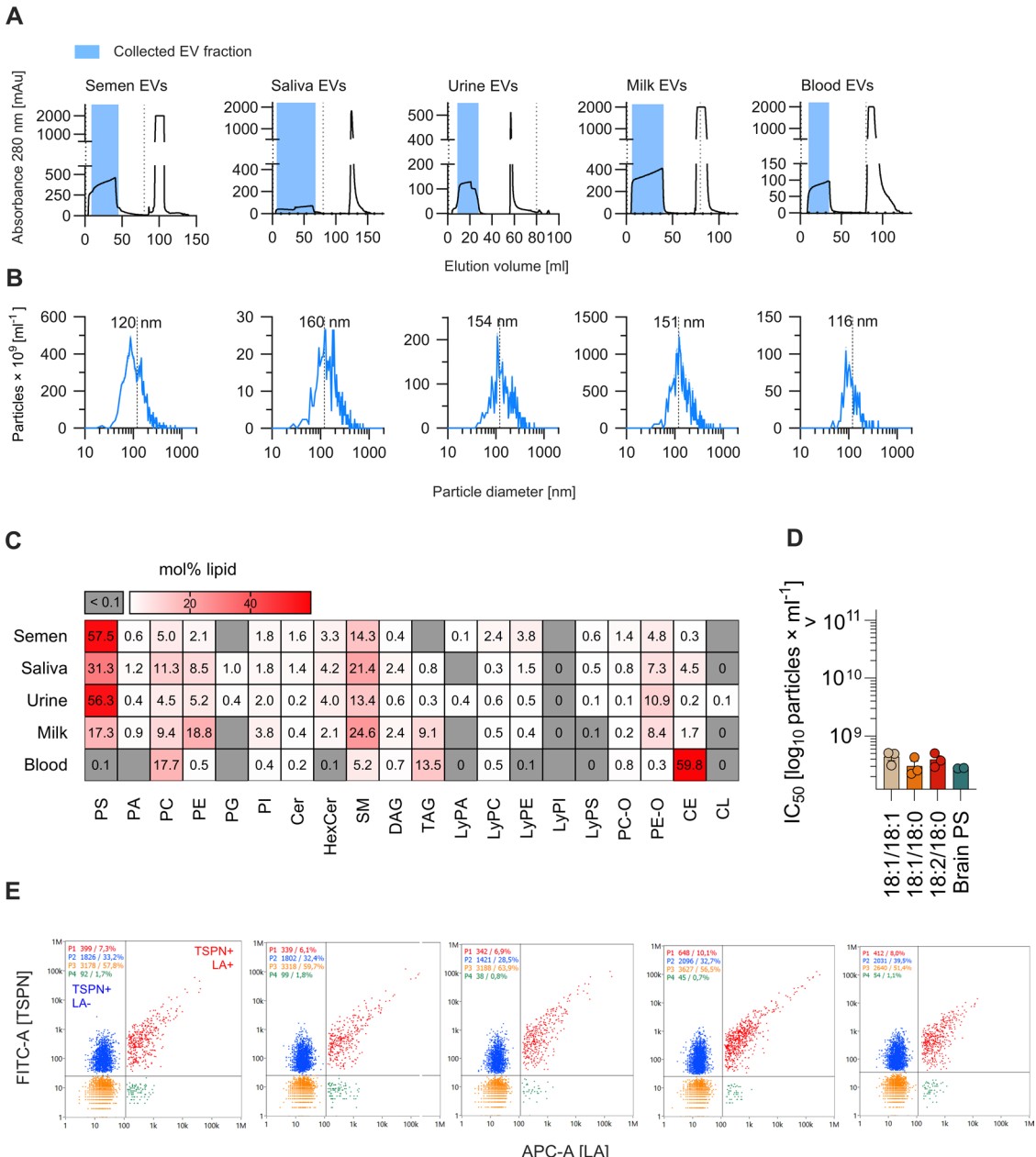

**Extended Data Fig. 3 | EV purification, characterization, and lipidomics details.** (**a**) Representative FPLC chromatograms showing A280 absorbance throughout SEC process; the collected EV fraction is shown in blue. (**b**) Representative NTA histograms showing size distribution detected in respective EV purifications from different sources. Lines and insets indicate mean particle diameter of 5 replicate acquisitions. (**c**) Phospholipid distribution in purified EVs from indicated sources analyzed by shotgun lipidomics. Shown are means of 2 biological replicates with averaged technical duplicates (saliva EVs) or 2-3 technical replicates of an individual EV preparation

(other sources), each from body fluids pooled from at least 35 (semen), 9 (saliva), 6 (urine), an individual (milk) or 3 donors (blood). See Supplementary Table 3 for all lipidomics data. (**d**) Half-maximal inhibitory concentration ($IC_{50}$) (ZIKV-MR766 MOI 0.25 on Vero E6) of liposomes containing 30 mol% PS with different individual lipid species or a biological mixture (brain PS). n = 3 independent experiments in biological triplicates, mean ± SD. (**e**) Nanoflow cytometry dot plots of five separate semen EV preparations stained for tetraspanins CD9/63/81 (TSPN) or PS (LA). Highlighted TSPN+ quadrants were used for quantification in Fig. 3i.

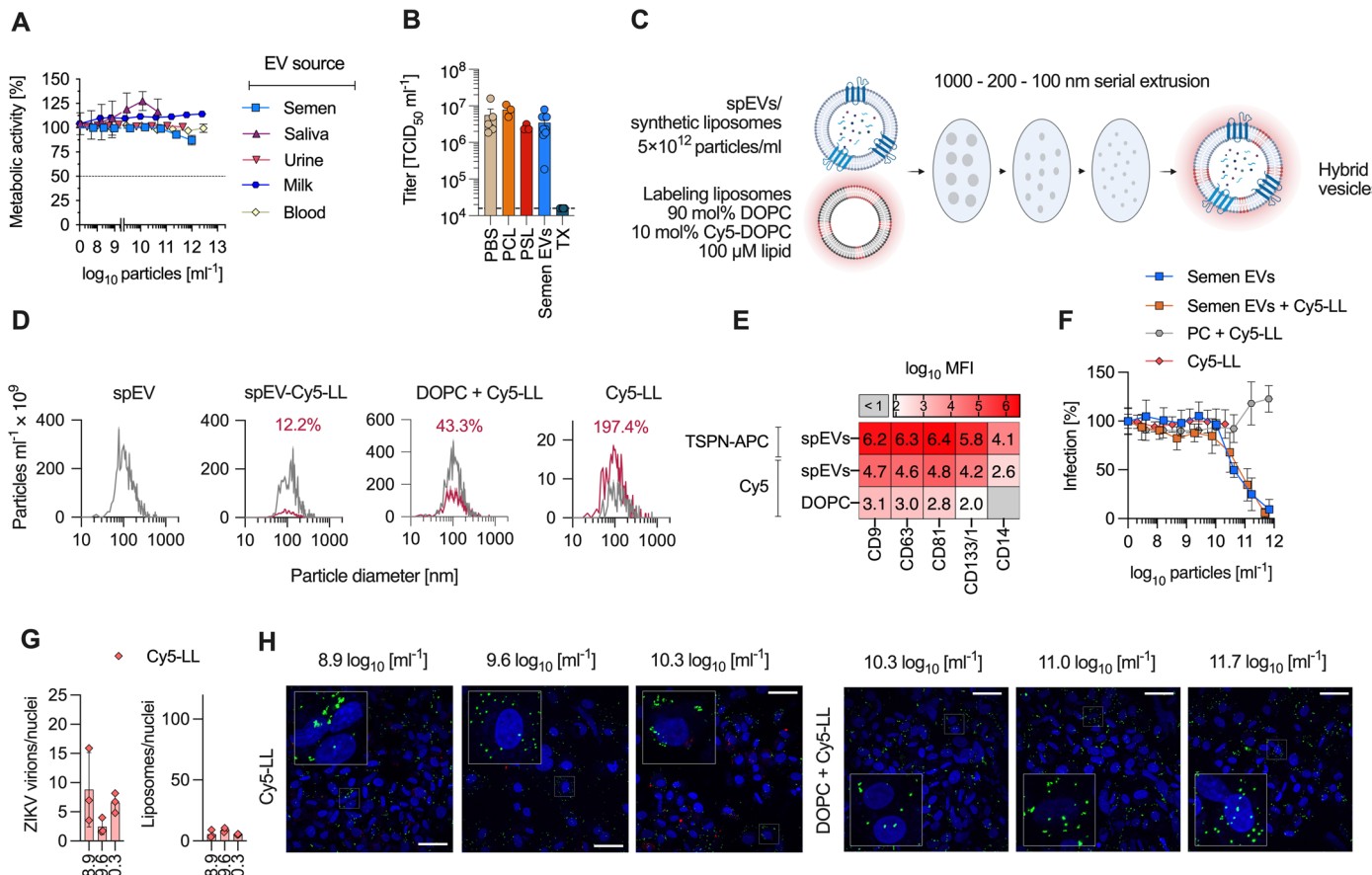

**Extended Data Fig. 4 | Cytotoxicity of EV preparations and preparation and characterization of EV-liposome hybrids.** (**a**) Effect of purified EVs on cellular metabolic activity. EVs were titrated onto Vero E6 cells and incubated with mock virus (medium only). Medium was replaced after 2 h and 2 days later, intracellular ATP levels were determined via CellTiter-Glo luminescent viability assay. Values normalized to untreated cells. n = 3 biological replicates, mean ± SD. (**b**) Evaluation of direct virucidal effects of semen EVs or liposomes. Virus stocks were treated with vesicles ($1 \times 10^{12}$ particles/ml) or 0.1% Triton X-100, TX, as a positive control for 2 h before determining residual infectious titer by $TCID_{50}$ endpoint titration on Vero E6 and evaluation of CPE 7 days post-infection. One-way ANOVA with Bonferroni's post-test, all without indication p > 0.05 = ns compared to PBS treated. n = 3 (liposomes), n = 5 (PBS), n = 6 (Triton X-100) and n = 9 (semen EVs) independently treated samples. Mean ± SEM. (**c**) Workflow for

preparation of hybrid EV-liposome vesicles. (**d**) Representative NTA histograms showing size distribution detected in scatter (grey) or 640 nm fluorescence mode (red), n = 3 replicate acquisitions, shaded area show SD. (E) Bead-assisted flow cytometric analysis of EV surface proteins by TSPN staining or detecting Cy5 with PC liposomes as control. n = 3 replicates (**f**) Antiviral activity (ZIKV MR766 MOI 0.2 on Vero E6) of unlabeled semen EVs, labeled semen EVs and liposomes, as well as labeling-liposomes. n = 3 biological replicates, means ± SD. (**g**) Attachment of virions (left) or vesicles (right) after addition of labeling-liposomes and ZIKV MR766 (MOI 35) to Vero E6 cells. Quantification of 3 separate image per condition (n = 3), means ± SD. (**h**) Representative confocal microscopy images as used for quantification shown in (F) including images for or DOPC liposomes mixed with labeling-liposomes (shown in Fig. 4c). Scale bars = 20 μm.

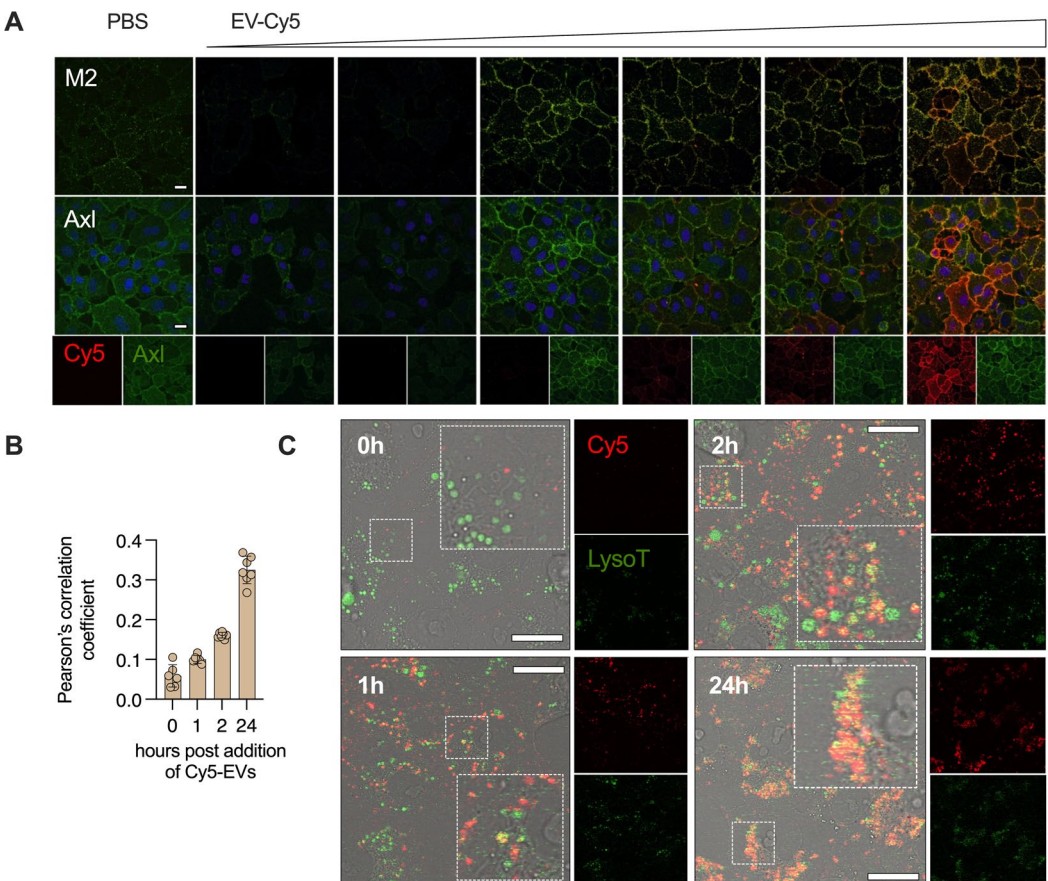

**Extended Data Fig. 5 | EVs bind Axl and are uptaken in lysosomes. (a)** Receptor binding of hybrid EVs. Cy5-labled hybrid semen EVs (shown in red) were added to Vero E6 cells in 8-well ibidi slides at increasing concentrations and slides incubated at 4 °C for 2 h to allow attachment without uptake. Cells were then fixed by addition of 4% PFA and additionally stained for Axl (anti-Axl-PE, Thermo Fisher, #12-1087-42, shown in green) and nuclei (Hoechst, shown in blue). Images were acquired as z-stacks, maximum intensity projections are shown as the pearson's correlation map, merge and single channels. Mander's M2 overlap maps were calculated and visualized from z-stacks using Huygens Professional.

Images representative of 3 images per concentration. Scale bars = 20 μm. (B-C) Cellular uptake of Cy5-semen EVs on Vero E6 cells. Cells were seeded to an Ibidi chamber one day prior, then stained with LysoTracker DND-26 Green (LysoT, shown in green) before adding Cy5-hybrid EVs (5 × 10^{10} particles per ml, shown in red) and acquiring z-stacks at indicated time points, the imaging chamber being incubated at standard conditions between time points. Pearson's correlation coefficient (**b**) for correlation of Cy5 and LysoTracker channels calculated using LasX software. 6-7 separate z-stacks per time point, means ± SD. Images shown in (**c**) are representative snapshots of 6-7 total images acquired, scale bars = 20 μm.

# Reporting Summary

## Statistics

For all statistical analyses, confirm that the following items are present in the figure legend, table legend, main text, or Methods section.

| n/a | Confirmed | |
|---|---|---|
| ☐ | ☒ | The exact sample size (*n*) for each experimental group/condition, given as a discrete number and unit of measurement |
| ☐ | ☒ | A statement on whether measurements were taken from distinct samples or whether the same sample was measured repeatedly |
| ☐ | ☒ | The statistical test(s) used AND whether they are one- or two-sided<br>*Only common tests should be described solely by name; describe more complex techniques in the Methods section.* |
| ☐ | ☒ | A description of all covariates tested |
| ☐ | ☒ | A description of any assumptions or corrections, such as tests of normality and adjustment for multiple comparisons |
| ☐ | ☒ | A full description of the statistical parameters including central tendency (e.g. means) or other basic estimates (e.g. regression coefficient) AND variation (e.g. standard deviation) or associated estimates of uncertainty (e.g. confidence intervals) |
| ☐ | ☒ | For null hypothesis testing, the test statistic (e.g. *F*, *t*, *r*) with confidence intervals, effect sizes, degrees of freedom and *P* value noted<br>*Give P values as exact values whenever suitable.* |
| ☒ | ☐ | For Bayesian analysis, information on the choice of priors and Markov chain Monte Carlo settings |
| ☐ | ☒ | For hierarchical and complex designs, identification of the appropriate level for tests and full reporting of outcomes |
| ☐ | ☒ | Estimates of effect sizes (e.g. Cohen's *d*, Pearson's *r*), indicating how they were calculated |

*Our web collection on statistics for biologists contains articles on many of the points above.*

## Software and code

Policy information about availability of computer code

**Data collection**

Luminescence data (Cell-Titer Glo Luminescent viability assay, HIV-1 and HSV β-gal assay) was acquired using Simplicity 4.2 (Berthold Detection Systems). Absorbance-based assay data (viral protein immunodetection, BCA assay) was acquired using SoftMax Pro 7.0.3 (Molecular Devices). Flow cytometry data (Bead-assisted flow cytometry, HSV-1/2 infection assay) was acquired using CytExpert 2.3 (Beckman Coulter). Nanoparticle tracking data was acquired using ZetaVIEW using ZetaView Analyze 08.05.05 SP2 and 8.05.12 SP1. qPCR data (vaginal tissue block ZIKV replication) was acquired using StepOne Software 2.3. Confocal microscopy was performed using by confocal microscopy using a Zeiss LSM 710 and ZEN 2.3 (blue edition) software. DSB-3 reactions were recorded using a microplate fluorescence reader (BioTek Synergy) with Gen5 3.08.01. Nanoparticle flow cytometry was performed NanoAnalyzer (NanoFCM Co., Ltd, Nottingham, UK) and data collected using NanoFCM software (NF Profession V2.0). infected primary cells were imaged using Biotek Cytation with Gen5 Image+ 3.04.17. EVs preparation using ÄKTAstart chromatography system (Cytiva) was determined using A280 absorbance and UNICORN start 1.2.0.164.

**Data analysis**

Data was generally analyzed using Microsoft Excel for Mac, Version 16.71 and analyzed/visualized using GraphPad Prism for macOS Version 9.5.1. Statistical analyses were also performed using GraphPad Prism for macOS Version 9.5.1 and 10.0.2. For analysis of confocal microscopy data (attachment assay), Fiji (ImageJ) Version 2.3.0 was used with a macro that automatically identifies and counts local fluorescence maxima of a set of 1024 × 1024 px confocal images. Colocalization was evaluated using Huygens Professional software (Version 19.10) using Gaussian minimums for thresholding. FPLC chromatograms were analyzed using UNICORN start 1.2.0.164. (Cytiva). Flow cytometry data (Bead-assisted flow cytometry, HSV-1/2 infection assay) was analyzed using CytExpert 2.3 (Beckman Coulter). Nanoparticle tracking data was analyzed using ZetaVIEW Analyze 08.05.05 SP2 and 8.05.12 SP1. qPCR data (vaginal tissue block ZIKV replication) was analyzed using StepOne Software 2.3. NanoFCM data was evaluated by NanoFCM software (NF Profession V2.0). Illustrations created with BioRender.com.

For manuscripts utilizing custom algorithms or software that are central to the research but not yet described in published literature, software must be made available to editors and reviewers. We strongly encourage code deposition in a community repository (e.g. GitHub). See the Nature Portfolio guidelines for submitting code & software for further information.

## Data

Policy information about availability of data

All manuscripts must include a data availability statement. This statement should provide the following information, where applicable:

- Accession codes, unique identifiers, or web links for publicly available datasets
- A description of any restrictions on data availability
- For clinical datasets or third party data, please ensure that the statement adheres to our policy

The data generated during this study are provided as Source Data and Supplementary Code. No original code was used. Any additional information required is available from the lead contact upon request.

## Research involving human participants, their data, or biological material

Policy information about studies with human participants or human data. See also policy information about sex, gender (identity/presentation), and sexual orientation and race, ethnicity and racism.

| | |
|---|---|
| Reporting on sex and gender | Extracellular vesicles used in this study were isolated from pooled body fluids of healthy human donors. For saliva and urine, donors included both male and female sex donors; gender data was not collected. Semen was obtained as excess material from volunteer donors at a fertility center, gender data was not determined. Breast milk was obtained from female sex nursing mothers, gender was not inquired. Blood (serum) was obtained from a commercial source, which was collected from male sex donors; gender was not inquired. Data was not analyzed for gender or sex-specific effects, as the described phenomenon is assumed to be sex-independent based on previous data (Conzelmann C, Groß R, Zou M, Krüger F, Görgens A, Gustafsson MO, El Andaloussi S, Münch J, Müller JA. Salivary extracellular vesicles inhibit Zika virus but not SARS-CoV-2 infection. J Extracell Vesicles. 2020 Aug 24;9(1):1808281. doi: 10.1080/20013078.2020.1808281. PMID: 32939236; PMCID: PMC7480612). Vaginal tissue used for ex vivo infection studies was obtained as excess material from patients undergoing surgical procedures for treatment of vaginal prolapse; gender was not inquired. |
| Reporting on race, ethnicity, or other socially relevant groupings | Extracellular vesicles used in this study were isolated from pooled body fluids of healthy human donors. Race, ethnicity, or other socially grouping was not inquired. Vaginal tissue used for ex vivo infection studies was obtained as excess material from patients undergoing surgical procedures for treatment of vaginal prolapse; Race, ethnicity, or other socially grouping was not inquired. |
| Population characteristics | All samples used for extracellular vesicle isolation and vaginal tissue for infection were provided in an anonymized fashion and no data on age, genotypic information or past/current diagnosis or treatment of any disease was recorded. |
| Recruitment | Semen was obtained as excess material from volunteer donors at a fertility center, which may introduce a bias towards men with infertility. As the described phenonenon was donor independent in a previous study (Müller JA et al. Semen inhibits Zika virus infection of cells and tissues from the anogenital region. Nat Commun. 2018 Jun 7;9(1):2207. doi: 10.1038/s41467-018-04442-y. PMID: 29880824; PMCID: PMC5992203), we do not expect this to be be a confounding factor. Saliva and urine was collected from healthy, volunteering hospital employees. Breast milk was collected from volunteering nursing mothers. Participants were recruited via institution internal channels. There might be a bias towards german ethnicity, which we consider unlikely to impact the study outcome. Blood was obtained from a commercial source and no data on recruitment is available. Vaginal tissue was obtained as excess material from patients undergoing surgical procedures for treatment of vaginal prolapse with no selection bias other than the indication for surgery. |
| Ethics oversight | All experiments using human material were approved by the Ethics Comittee of Ulm University (Decision 88/17, 89/16, 89/17, 337/18, 131/16) |

Note that full information on the approval of the study protocol must also be provided in the manuscript.

# Field-specific reporting

Please select the one below that is the best fit for your research. If you are not sure, read the appropriate sections before making your selection.

☒ Life sciences  ☐ Behavioural & social sciences  ☐ Ecological, evolutionary & environmental sciences

For a reference copy of the document with all sections, see nature.com/documents/nr-reporting-summary-flat.pdf

# Life sciences study design

All studies must disclose on these points even when the disclosure is negative.

| | |
|---|---|
| Sample size | No statistical methods were used to calculate sample size for our biological experiments. Sample size was limited by availability of donor material (e.g. pooled body fluids necessary to yield sufficient concentrations of extracellular vesicles for downstream analysis). |
| Data exclusions | We did not exclude data from analysis |
| Replication | Experiments were performed with at least 3 biological replicates. Lipidomics analysis of EVs purified from pooled donors fluids was done with 2-3 technical replicates. Primary sample numbers dependent on the sample (n=2 for vaginal tissue; n=3-100 for most extracellular vesicle |

sources, see Table S2). Where limited by availability of donor material, only one biological replicate was analyzed (breast milk). All replication attempts were successful.

Randomization    Randomization was not applicable as the study solely involved in vitro and ex vivo experiments, where controlled conditions negated the need for random assignment typical in clinical trials.

Blinding    Blinding was not performed due to the nature of the study involving only in vitro and ex vivo experiments, without clinical treatments.

# Reporting for specific materials, systems and methods

We require information from authors about some types of materials, experimental systems and methods used in many studies. Here, indicate whether each material, system or method listed is relevant to your study. If you are not sure if a list item applies to your research, read the appropriate section before selecting a response.

## Materials & experimental systems

| n/a | Involved in the study |
|---|---|
| ☐ | ☒ Antibodies |
| ☐ | ☒ Eukaryotic cell lines |
| ☒ | ☐ Palaeontology and archaeology |
| ☒ | ☐ Animals and other organisms |
| ☒ | ☐ Clinical data |
| ☒ | ☐ Dual use research of concern |
| ☒ | ☐ Plants |

## Methods

| n/a | Involved in the study |
|---|---|
| ☒ | ☐ ChIP-seq |
| ☐ | ☒ Flow cytometry |
| ☒ | ☐ MRI-based neuroimaging |

## Antibodies

Antibodies used

Anti-Flavivirus group antigen [D1-4G2-4-15 (4G2)]        Absolute Antibody        Cat. # Ab00230-2.0
RRID:AB_2715504 (1:5,000 for WNV in-cell ELISA; 1;10,000 for ZIKV in-cell ELISA and confocal microscopy)

CD9 (D8O1A) Rabbit mAb 13174  Cell Signaling Technology        Cat. # 13174
RRID: AB_2798139 (1:1,000 in western blotting)

Flotillin-1 (D2V7J) XP® Rabbit mAb 18634      Cell Signaling Technology        Cat. # 18634
RRID: AB_2773040 (1:1,000 in western blotting)

Goat anti-Mouse IgG (H+L) Secondary Antibody, HRP      Thermo Fisher Scientific        Cat. # 31430
RRID: AB_228307 (1:10,000 for ZIKV and WNV in-cell ELISA; 1:15,000 for SARS-CoV-2 in-cell ELISA)

Goat anti-mouse (H+L) Alexa Fluor 488        Thermo Fisher Scientific        Cat. # A-11001
RRID: AB_2534069 (1:400 in ZIKV immunoimaging)

Goat anti-Rabbit Affinity HRP      Thermo Fisher Scientific        Cat. # 31460
RRID: AB_228341 (1:10,000 in western blotting)

Goat anti-mouse IgG (H+L) Superclonal Recombinant Secondary AB Alexa Fluor 647        Thermo Fisher Scientific        Cat. # A27029
RRID: AB_2536092  (1:1,000 in ZIKV attachment imaging)

Goat anti-rabbit IgG StarBright 520        Bio-Rad      Cat. # 12005870
RRID: AB_2884949 (1:2,500 in western blotting)

Goat anti-rabbit IgG StarBright 700        Bio-Rad      Cat. # 12004162
RRID: AB_2721073 (1:2,500 in western blotting)

HSP70 (D69) Antibody 4876        Cell Signaling Technology        Cat. # 4876
RRID: AB_2119693  (1:1,000 in western blotting)

SARS-CoV/SARS-CoV-2 N 40143-MM05 Mouse mAb        SinoBiological        Cat. # 40143-MM05
RRID: AB_2827977 (1:5,000 in SARS-CoV-2 in-cell ELISA)

Axl Monoclonal Antibody (DS7HAXL), PE        Thermo Fisher Scientific        Cat. 12-1087-42
RRID: AB_2723961 (120 ng/ml in confocal microscopy)

CD9 (HI9a) FITC mouse mAb 312104, Biolegend, Cat. # 312104
RRID: AB_2075894 (12.5 μM in nanoflow cytometry)

CD81 (TAPA-1) FITC mouse mAB 349504, Biolegend, Cat. # 349504
RRID: AB_2075894 (12.5 μM in nanoflow cytometry)

CD63 (H5C6) FITC mouse mAB 353005, Biolegend, Cat. # 353005
RRID: AB_10898319 (12.5 µM in nanoflow cytometry)

IgG1, κ (MOPC-21) mouse mAB 400109, Biolegend, Cat. # 400109
RRID: AB_2861401 (12.5 µM in nanoflow cytometry)

IgG2a, κ (MOPC-173) mouse mAB 400207, Biolegend, Cat. # 400207
RRID: AB_2884007 (12.5 µM in nanoflow cytometry)

| Validation | Antibodies were validated by the manufacturers. |

## Eukaryotic cell lines

Policy information about cell lines and Sex and Gender in Research

| Cell line source(s) | BHK-21 - obtained from ATCC Cat. # CCL-10<br>Calu-3 - obtained from M. Frick, Ulm University (ATCC # HTB-55)<br>HEK293T - obtained from ATCC Cat. # CRL3216<br>HFF - primary, obtained from J. von Einem, Ulm University Medical Center<br>TZM-bl (HeLa derived) - obtained from NIBSC Cat. # ARP5011<br>Vero E6 - obtained from ATCC Cat. # CRL-1586<br>Huh-7 - obtained from R. Bartenschlager, Heidelberg University, described in Windisch et al., 2005 and Lohmann and Bartenschlager 2014<br>ELVIS - obtained from ATCC described in Profitt et al, 1995 |
| Authentication | Cells were purchased from indicated companies or labs and used without further authentication. |
| Mycoplasma contamination | All cells were regularly tested for mycoplasma contamination and tested negative. |
| Commonly misidentified lines<br>(See ICLAC register) | No commonly misidentified cell lines were used. |

## Plants

| Seed stocks | *Report on the source of all seed stocks or other plant material used. If applicable, state the seed stock centre and catalogue number. If plant specimens were collected from the field, describe the collection location, date and sampling procedures.* |
| Novel plant genotypes | *Describe the methods by which all novel plant genotypes were produced. This includes those generated by transgenic approaches, gene editing, chemical/radiation-based mutagenesis and hybridization. For transgenic lines, describe the transformation method, the number of independent lines analyzed and the generation upon which experiments were performed. For gene-edited lines, describe the editor used, the endogenous sequence targeted for editing, the targeting guide RNA sequence (if applicable) and how the editor was applied.* |
| Authentication | *Describe any authentication procedures for each seed stock used or novel genotype generated. Describe any experiments used to assess the effect of a mutation and, where applicable, how potential secondary effects (e.g. second site T-DNA insertions, mosiacism, off-target gene editing) were examined.* |

## Flow Cytometry

### Plots

Confirm that:

☒ The axis labels state the marker and fluorochrome used (e.g. CD4-FITC).

☒ The axis scales are clearly visible. Include numbers along axes only for bottom left plot of group (a 'group' is an analysis of identical markers).

☒ All plots are contour plots with outliers or pseudocolor plots.

☒ A numerical value for number of cells or percentage (with statistics) is provided.

### Methodology

| Sample preparation | For flow cytometric determination of HSV-1 and -2 infection by virus-encoded eGFP, cells (Vero E6) were trypsinized (0.25% Trypsin), transferred to V-well plates, centrifuged and fixed in 4% PFA (in PBS) for 1h at 4°C. Fixed cells were then acquired by flow cytometry.<br><br>For determination of EV surface antigens using bead-assisted flow cytometry, the MACSPlex Exosome Kit, human (Milteny Biotec) was used as described by the manufacturer and bead-bound EVs stained with the included anti-Tetraspanin cocktail or Alexa 647-conjugated bovine Lactaderine (BLAC-A647, Haematologic Technologies). |

For nano-flow cytometry, 12.5 μM of corresponding antibodies in 50 μl TE or 5 μg/ml lactadherin-A647 was used. After removing antibody aggregates by centrifugation at 12,000×g for 10 min, the supernatant was added to 2 × 10^8 particles, followed by incubation for 12 h at 37∘C under constant shaking and washing with 1 ml 1x TE buffer by ultracentrifugation at 110,000×g for 45 min at 4∘C (Beckman Coulter MAX-XP centrifuge, TLA-45 rotor; Beckman Coulter, Krefeld, Germany). The pellet was resuspended in 50 μl 1x TE buffer for nFCM analysis. Monodisperse silica beads (NanoFCM Co., Ltd, Nottingham, UK) of four different diameters (68 nm; 91 nm; 113 nm; 155 nm) served as size reference standards. Freshly filtered (0.1 μm) 1x TE buffer pH 7.4 was analyzed to define the background signal, which was subtracted from all other measurements. The EV samples were diluted with filtered (0.1 μm) 1x TE buffer, resulting in a particle count in the optimal range of 2,500–12,000 events.

| Instrument | Cytoflex LX, Beckman Coulter; Flow NanoAnalyzer , NanoFCM |

| Software | CytExpert 2.3; NF Profession V2.0 |

| Cell population abundance | For analysis of cell lines and beads purity is not applicable. Extracellular vesicles preparation is thoroughly described in the manuscript. |

| Gating strategy | Single cells and beads were gated based on FSC/SSC plots. The gating strategy for MACSplex beads was previously shown (Wiklander OPB, Bostancioglu RB, Welsh JA, Zickler AM, Murke F, Corso G, Felldin U, Hagey DW, Evertsson B, Liang XM, Gustafsson MO, Mohammad DK, Wiek C, Hanenberg H, Bremer M, Gupta D, Björnstedt M, Giebel B, Nordin JZ, Jones JC, El Andaloussi S, Görgens A. Systematic Methodological Evaluation of a Multiplex Bead-Based Flow Cytometry Assay for Detection of Extracellular Vesicle Surface Signatures. Front Immunol. 2018 Jun 13;9:1326. doi: 10.3389/fimmu.2018.01326. PMID: 29951064; PMCID: PMC6008374.) No gating was performed for nano-flow cytometry, all events are shown. |

☒ Tick this box to confirm that a figure exemplifying the gating strategy is provided in the Supplementary Information.

