## [Peer Review File · Nature Microbiology]

Peer Review Information

Journal: Nature Microbiology

Manuscript Title: Phosphatidylserine-exposing extracellular vesicles in body fluids inhibit viral apoptotic mimicry infection

Corresponding author name(s): Janis Muller

Reviewer Comments & Decisions:

Decision Letter, initial version:

Message 26th May 2023

:

Dear Janis,

Thank you and your co-authors for answering my previous e-mail and for your patience while your manuscript "Phosphatidylserine-exposing extracellular vesicles in body fluids inhibit viral apoptotic mimicry infection" was under peer-review at Nature Microbiology. It has now been seen by 3 referees, whose expertise and comments you will find at the end of this email. Although they find your work of some potential interest, they have raised a number of concerns that will need to be addressed before we can consider publication of the work in Nature Microbiology.

In particular, referee #1 requests to show the raw data of your flow cytometry analysis, to investigate whether lipid composition changes after time and what the molecular forms of PS are in different bodily fluids. Referee #2 suggests to stain natural EVs which are isolated from body fluids and asks about the turnover rate of EVs, as well as whether EVs bind to and sequester virions. We are overruling the novelty concerns raised by referee #3.

Should further experimental data allow you to address these criticisms, we would be happy to look at a revised manuscript.

We strongly support public availability of data. Please place the data used in your paper into a public data repository, if one exists, or alternatively, present the data as Source Data or Supplementary Information. If data can only be shared on request, please explain

why in your Data Availability Statement, and also in the correspondence with your editor. For some data types, deposition in a public repository is mandatory - more information on our data deposition policies and available repositories can be found at <https://www.nature.com/nature-research/editorial-policies/reporting-standards#availability-of-data>.

Please include a data availability statement as a separate section after Methods but before references, under the heading "Data Availability". This section should inform readers about the availability of the data used to support the conclusions of your study. This information includes accession codes to public repositories (data banks for protein, DNA or RNA sequences, microarray, proteomics data etc...), references to source data published alongside the paper, unique identifiers such as URLs to data repository entries, or data set DOIs, and any other statement about data availability. At a minimum, you should include the following statement: "The data that support the findings of this study are available from the corresponding author upon request", mentioning any restrictions on availability. If DOIs are provided, we also strongly encourage including these in the Reference list (authors, title, publisher (repository name), identifier, year). For more guidance on how to write this section please see: <http://www.nature.com/authors/policies/data/data-availability-statements-data-citations.pdf>

* If you have not done so already we suggest that you begin to revise your manuscript so that it conforms to our Article format instructions at <http://www.nature.com/nmicrobiol/info/final-submission>. Refer also to any guidelines provided in this letter.

When submitting the revised version of your manuscript, please pay close attention to our [href="https://www.nature.com/nature-portfolio/editorial-policies/image-integrity">Digital Image Integrity Guidelines](https://www.nature.com/nature-portfolio/editorial-policies/image-integrity). and to the following points below:

- that unprocessed scans are clearly labelled and match the gels and western blots presented in figures.
- that control panels for gels and western blots are appropriately described as loading on sample processing controls
- all images in the paper are checked for duplication of panels and for splicing of gel

2lanes.

Note: This url links to your confidential homepage and associated information about manuscripts you may have submitted or be reviewing for us. If you wish to forward this e-mail to co-authors, please delete this link to your homepage first.

Nature Microbiology is committed to improving transparency in authorship. As part of our efforts in this direction, we are now requesting that all authors identified as 'corresponding author' on published papers create and link their Open Researcher and Contributor Identifier (ORCID) with their account on the Manuscript Tracking System (MTS), prior to acceptance. This applies to primary research papers only. ORCID helps the scientific community achieve unambiguous attribution of all scholarly contributions. You can create and link your ORCID from the home page of the MTS by clicking on 'Modify my Springer Nature account'. For more information please visit please visit www.springernature.com/orcid.

If you wish to submit a suitably revised manuscript we would hope to receive it within 6 months. If you cannot send it within this time, please let us know. We will be happy to consider your revision, even if a similar study has been accepted for publication at Nature Microbiology or published elsewhere (up to a maximum of 6 months).

Yours sincerely,

Reviewer Expertise:

Referee #1: Lipidomics
Referee #2: EVs in pathogenesis
Referee #3: Arboviruses

Reviewer Comments:

Reviewer #1 (Remarks to the Author):

Surface exposure of PS is emblematic "eat-me" signal that tags cells for efferocytosis. However not all externalized PS appears to be a functionally equivalent. Although surface exposed PS functions as a dominant and evolutionarily conserved innate

immunosuppressive signal that tags cells for efferocytosis, PS is also externalized on extracellular vesicles (EVs), on the surface of exosomes, participates in various signaling events such as membrane association and activation of Akt and PKC etc. Moreover, an innate immunosuppressive effect of externalized PS can be hijacked by numerous viruses to promote infection or establish infection latency. As obligate intracellular pathogens, viruses employ an elegant strategies such as viral apoptotic mimicry, hijacking plasma or intracellular membranes and externalizing PS to evade host response and assure their successful binding, entry, replication and immune evasion.

Authors of submitted MS demonstrate that different secreted body fluids contain large numbers of EVs that externalize PS. Authors successfully employed a targeted, enzymatic reaction to eliminate PS primarily in the outer leaflet of virions (Fig. 1B) and performed a set of independent experiments to verify whether intact and "shaved" EVs and virions compete for viral apoptotic mimicry cell attachment and prevent an infection by the same mechanism as administered PS-liposomes.

Authors anticipate an existence of very attractive mechanism which can provide the cells with an emergency innate defense surveillance mechanism until other (but slower) antiviral innate defense mechanisms are put in place. Surface exposing PS EVs (Fig. 3H) or PS-liposomal mimetics (Fig. 1C) but not head group soluble (Figure 2C) or aminophospholipid (PE) counterparts (Fig. 2D) prevent in competitive manner (extended Data Fig 4) both attachment (Fig. 2B, Figs 2F-H) and infection in dose-dependent manner (Figs. 1B and 1D) of viruses using particularly an apoptotic mimicry mechanism for infection (Fig. 1B and Fig. 5).

This is a very technically sound and thoroughly executed experimentally study. I am glad to see that effect of PE-liposomal mimetics was addressed. The experimental aspects of this work are of very good quality and the manuscript is written well and appropriately concise. All cited manuscripts are properly referenced and cited.

Nevertheless I have several concerns which should be addressed by authors:

1. What are specialized molecular forms (acyl length, unsaturation level) of PS in EVs from different body fluids?

2. Fig. 1B. What is the lipid composition of EVs after a 2-h treatment with PSDC?

What is decarboxylation limit of this enzyme? As has been seen before, beyond 5 % PSDC does not further decarboxylate PS due to a minimal electrostatic attraction.

To track the degree of progress of the PSDC reaction, authors can follow the production of PE by measurement of fluorescence yield ($\lambda_{ex} = 403 \text{ nm}$, $\lambda_{em} = 508 \text{ nm}$) of the distyrylbenzene-bis-aldehyde (DSB-3)-PE adducts occurred after the primary amine reacted with this water-soluble fluorophore and provided strong discrimination against the PS substrate. (Choi et al., A novel fluorescence assay for measuring phosphatidylserine decarboxylase catalysis. *J Biol Chem.* 2018 Feb 2;293(5):1493-1503)

3. Lane 709. PS decarboxylation. This section in Methods should be described in details. Did Authors use a recombinant version of a water soluble His6-tagged PS decarboxylase from parasite *Plasmodium knowlesi* (PkPSD), which was utilized to selectively decarboxylate PS in the outer leaflet of EVs, converting it to PE?

Authors said "before 10-fold dilution of the mixtures on cells". Authors should rephrase and clarify this sentence.

After 2-h reaction, the PSDC should be removed. Whether 10X dilution PSD was utilized to "remove" PSDC before inoculation?

If recombinant PSDC is His- tagged, the presence of a tag can facilitate PSDC removal using Ni-NTA agarose beads . Affinity adsorption and subsequent centrifugation can provide a gentle method for separating PSDC from EVs.

4. Fig. 2. What is critical concentration or topography of PS need to be acquired for interference with virion attachment?

5. A symmetry PS in EVs is not absolute. Decarboxylation of PS can lead to an exposure of luminal PS molecules from the inner to the outer leaflet of EVs. After removal of the PSD, a flop of PS from the inner to the outer leaflet would reestablish PS asymmetry. It is important to verify the membrane asymmetry of the isolated vesicles and to know how long it is retained.

6. Figure 3H. A targeted bead-assisted flow cytometry was used identify and quantitate PS positive and negative sub-populations of EVs. It would be very interesting to know the exact percentage of EVs that presumably expose PS. Such information can be obtained from the binding of surface exposed PS to Ca⁺⁺-independent fluorophore-conjugated lactadherin lactadherin-A647. Representative raw data with flow cytometry quadrants or histograms should be provided in order to support a heat map shown on Fig. 3H (ESSENTIAL).

7. Cyclodextrin-mediated phospholipid exchange between PS liposomes and EVs can be utilized to supply PS back to EVs after PSDC treatment in order to keep the same protein-lipid background of different EVs to see whether these EVs retain the essentially the same ability to compete for viral apoptotic mimicry cell attachment and prevent an infection. (Li et al., Replacing plasma membrane outer leaflet lipids with exogenous lipid without damaging membrane integrity. PLoS One. 2019 Oct 7;14(10):e0223572

Reviewer #2 (Remarks to the Author):

The work of Groß et al. reports on a new function of extracellular vesicles (EVs): Defending against infection with a wide group of viruses that carry PS and use PS receptor for interaction with target cells. These viruses include Zika, Dengue, West Nile, Chikungunya, Ebola, and VSV, but not HIV-1, SARS-Cov-2, HCV, HSV and several other viruses. According to the authors, EVs that carry PS are generated upon viral infection and compete with viruses for binding to the cell membrane PS. Since the number of such EVs are far larger than the number of virions, viral infection is suppressed. The authors provide convincing experimental data that support this concept.

In particular, they showed that liposomes containing PS suppress Zika infection, while liposomes that contain only PC do not. Also, PS-containing liposomes do not suppress infection by HIV or HSV that do not use PS receptors for infection. To inhibit viral infection PS-containing liposomes have to be present before or during viral inoculation indicating that they interfere with viral entry. Confocal images revealed that PS-containing liposomes are attached to the cell surface.

To bring their experimental system closer to the in vivo situation, the authors isolated EVs from pooled human semen, saliva, urine, breast milk, and blood. The amount of PS in these EVs varied depending on the body fluid they were isolated from. The highest concentration of PS was in semen, saliva, and urea, while the lowest was in blood. Significant amount of these EVs expose PS on their surface. Accordingly, hybrid EV-liposome vesicles and EVs from blood protect cells or cervical explants from ZIKV less efficiently than EVs from semen. There was no such effect when a panel of viruses that do not use PS receptors for viral entry. These results may explain the difference in viral infection pathways in nature.

In general, the results of the experiments seem to be convincing. Figures that accompany the text clearly support the conclusions.

My critical remarks are minor and do not challenge the authors' concept.

1. It would be interesting to know what fraction of cell membrane PS are occupied by PS-containing EVs at a particular concentration. This could have been evaluated in the course of the same experiments that the authors performed.

2. Another question is regarding the fate of the bound EVs. Using the mechanism of viral apoptotic mimicry, EVs should like viruses, enter the cells and decrease the amount of the surface-exposed PS. Do the authors have an idea of the timeline of this process and of the rate of the PS turnover?

3. How do the authors exclude direct interactions of EVs with virions that may lead to inactivation of the latter?

4. While the authors used hybrids of fluorescent liposomes and natural EVs to visualize these vesicles binding, why did not they use fluorescent dyes like Bodipy or similar to stain natural EVs isolated from body fluids?

5. Finally, when the authors present “representative” images, they should mention how many replicates these images in the Figures represent.

Reviewer #3 (Remarks to the Author):

The study by Grob et al. is a follow-up of a previous work published by the Muller’s team showing that semen and saliva contain extracellular vesicles (EVs) block Zika virus (ZIKV) infection both in vitro and in an ex-vivo model. Here, the authors have investigated the underlying mechanisms and showed that EVs from body fluids expose the lipid phosphatidylserine (PS) on their outer membrane layer. PS is also known to be exposed at the viral envelope of several viruses and is essential for their infectious entry. This mechanism, known as viral apoptotic mimicry, is inhibited by molecules that bind to PS with high affinity such as annexin A5. In this manuscript, the authors showed that PS-exposing EVs prevent infection of ZIKV and other enveloped viruses that expose PS at their surface such as dengue, chikungunya or Ebola viruses. In my opinion, this manuscript does not bring much new information when compared to what is already known in the literature. The experiments described in the Figure 1, 2 and 5 of the manuscript are very well-performed but unfortunately most of the results presented in these figures are expected and a repetition of data already published by others (PMID: 21501828, PMID: 26052667, PMID: 23555248, PMID: 23084921). The novelty of this manuscript is the idea that PS-exposing EVs in secreted fluids may act as an innate defense barrier that regulate the transmission routes of viruses that use apoptotic mimicry. Unfortunately, the data presented by the authors do not support this concept. Overall, given the experimental redundancy of this manuscript with already published papers, this study lacks novelty and does not provide a significant advance in our understanding of the role of EVs in modulating the transmission route of apoptotic mimicry viruses. In my opinion, this manuscript is more suitable to a specialized journal.

Author Rebuttal to Initial comments

Reply to the reviewers’ comments (in blue)

Reviewer #1: Surface exposure of PS is emblematic “eat-me” signal that tags cells for efferocytosis. However not all externalized PS appears to be a functionally equivalent. Although

7surface exposed PS functions as a dominant and evolutionarily conserved innate immunosuppressive signal that tags cells for efferocytosis, PS is also externalized on extracellular vesicles (EVs), on the surface of exosomes, participates in various signaling events such as membrane association and activation of Akt and PKC etc. Moreover, an innate immunosuppressive effect of externalized PS can be hijacked by numerous viruses to promote infection or establish infection latency. As obligate intracellular pathogens, viruses employ an elegant strategies such as viral apoptotic mimicry, hijacking plasma or intracellular membranes and externalizing PS to evade host response and assure their successful binding, entry, replication and immune evasion.

Authors of submitted MS demonstrate that different secreted body fluids contain large numbers of EVs that externalize PS. Authors successfully employed a targeted, enzymatic reaction to eliminate PS primarily in the outer leaflet of virions (Fig. 1B) and performed a set of independent experiments to verify whether intact and “shaved” EVs and virions compete for viral apoptotic mimicry cell attachment and prevent an infection by the same mechanism as administered PS-liposomes.

Authors anticipate an existence of very attractive mechanism which can provide the cells with an emergency innate defense surveillance mechanism until other (but slower) antiviral innate defense mechanisms are put in place. Surface exposing PS EVs (Fig. 3H) or PS-liposomal mimetics (Fig. 1C) but not head group soluble (Figure 2C) or aminophospholipid (PE) counterparts (Fig. 2D) prevent in competitive manner (extended Data Fig 4) both attachment (Fig. 2B, Figs 2F-H) and infection in dose-dependent manner (Figs. 1B and 1D) of viruses using particularly an apoptotic mimicry mechanism for infection (Fig. 1B and Fig. 5).

This is a very technically sound and thoroughly executed experimentally study. I am glad to see that effect of PE-liposomal mimetics was addressed. The experimental aspects of this work are of very good quality and the manuscript is written well and appropriately concise. All cited manuscripts are properly referenced and cited.

We appreciate the very positive feedback.

Nevertheless I have several concerns which should be addressed by authors:

1. What are specialized molecular forms (acyl length, unsaturation level) of PS in EVs from different body fluids?

To address this point, we now include information about the acyl length, saturation and hydroxylation state of PS and all other lipid species included in our lipidomics approach in the new Table S3 (lines 194-196). In addition, we determined the antiviral activities of liposomes consisting of 30 mol% PS with 18:0/18:1, 18:2/18:0, or 18:1/18:1 fatty acids, as well as a mixture of naturally occurring PS species from porcine brain tissue (primarily containing 18:0, 18:1 and 22:6 acyl chains) in a DOPC (18:1/18:1) background. There were no significant differences in antiviral activity (new Extended Data Figure 3E; lines 201-203).

2. Fig. 1B. What is the lipid composition of EVs after a 2-h treatment with PSDC? What is decarboxylation limit of this enzyme? As has been seen before, beyond 5 % PSDC does not further decarboxylate PS due to a minimal electrostatic attraction. To track the degree of progress of the PSDC reaction, authors can follow the production of PE by measurement of fluorescence yield ($\lambda_{\text{ex}} = 403 \text{ nm}$, $\lambda_{\text{em}} = 508 \text{ nm}$) of the distyrylbenzene-bis-aldehyde (DSB-3)-PE adducts occurred after the primary amine reacted with this water-soluble fluorophore and provided strong discrimination against the PS substrate. (Choi et al., A novel fluorescence assay for measuring phosphatidylserine decarboxylase catalysis. *J Biol Chem.* 2018 Feb 2;293(5):1493-1503)

We thank we reviewer for bringing DSB-3 to our attention – this compound enabled us to determine the reaction kinetics of PSDC. DSB-3 measurements show that decarboxylation viral particles is completed after 5 min (new Extended Data Figure 1A, lines 92-95).

Upon treatment of EVs with PSDC, we observed a rapid reaction that reaches a plateau after 15 min (Figure R1A) and detected incomplete conversion from PS to PE with thin-layer chromatography (Figure R1B). We then blocked possible residual PS using lactadherin (PSDC + Block) but did not observe a reduction in antiviral activity of semen EVs (Figure R1C-D). Likely, this can be attributed to a combination of factors: As also stated by the reviewer, soluble PSDC has a decarboxylation limit of ~ 6 mol% (Drechsler et al., *Biophys J*, 2018), which is still sufficient for potent antiviral activity (Figure 1C+D). Thus, even a hypothetically optimal reduction of 57.5 mol% PS in semen EVs (Extended Data Figure 3D) to 6 mol% PS would not

impair its antiviral activity. In addition, lactadherin used for blocking saturates at ~1 mol% PS (Otzen et al., *Biochim Biophys Acta*, 2012) and is thus ineffective against vesicles containing higher residual concentrations of PS. Finally, the cleavage product of PSDC is phosphatidylethanolamine, which has residual affinity to PS receptors (Figure 2D) and may even synergize with PS to enhance vesicle binding (Zhang et al., *J Virol*, 2020). Since PSDC does not allow to remove sufficient PS from EVs to reduce antiviral activity, we decided to use phospholipase D combined with a lactadherin block to deplete EVs from PS. This approach was successful and significantly reduced the antiviral activity of these EVs (new Figure 4F-I, lines 273-283).

Figure R1: PSDC digestion of EVs and blockade of residual PS by lactadherin is insufficient for reducing antiviral activity. (A) Kinetics of PSDC reaction of liposomes and purified semen EVs. Liposomes (containing 50 mol% PS or PE in a PC background) or semen EVs, prepared and purified in decarboxylation buffer, respectively, were treated with 50 µg/ml PSDC and incubated at 30°C. Samples were taken at indicated timepoints, reactions stopped by addition of sodium tetraborate buffer (pH 9). After termination of the kinetic experiment at 60 min, PS/PE conversion was quantified by changes in DSB-3 fluorescence. 1 (50% PS liposomes) or 2 (semen EVs, 50% PE liposomes) samples analyzed simultaneously, means ± SD shown. (B) Verification of PE generation from 50 mol% PS liposomes (after 1 h reaction with 50 µg/ml PSDC) using phospholipid HPTLC and iodine vapor stain. (C) Representative dataset (triplicates, means ± SEM) and IC₅₀ of 5 separately treated EV samples (D, means ± SEM) for antiviral activity of untreated control EVs and EVs treated with PSDC and additionally lactadherin to block residual PS (“PSDC + Block”). One-way ANOVA with Bonferroni’s post-test. ns = p > 0.05, not significant.

3. Lane 709. PS decarboxylation. This section in Methods should be described in details. Did Authors use a recombinant version of a water soluble His6-tagged PS decarboxylase from parasite *Plasmodium knowlesi* (PkPSD), which was utilized to selectively decarboxylate PS in the outer leaflet of EVs, converting it to PE?

We expanded the methods section to describe the process more clearly and clarified that we indeed used the His-tagged, recombinantly produced PS decarboxylase of *P. knowlesi* (Lines 787-790).

Authors said “before 10-fold dilution of the mixtures on cells”. Authors should rephrase and clarify this sentence.

We clarified this to clearly indicate that mixtures of virus and PSDC were added to cells at 10-fold dilution, resulting in up to 10 µg/ml PSDC on cells (Lines 799-802).

After 2-h reaction, the PSDC should be removed. Whether 10X dilution PSD was utilized to “remove” PSDC before inoculation? If recombinant PSDC is His- tagged, the presence of a tag can facilitate PSDC removal using Ni-NTA agarose beads . Affinity adsorption and subsequent centrifugation can provide a gentle method for separating PSDC from EVs.

The reviewer is correct, we diluted the enzyme-virus mixture to reduce on-cell concentrations and confirmed the absence of cytotoxic effects (Extended Data Figure 1B), but did not remove the enzyme. This is now clarified in lines 799-802.

We repeated these experiments to compare effects with or without removal of PSDC by Ni-NTA bead affinity as suggested by the reviewer and show the results in Figure R2. Under both conditions dose-dependent antiviral activity against ZIKV was maintained (Figure R2A). However, PSDC removal by Ni-NTA itself reduced viral infectivity, suggesting that also virus particles were removed, despite spiking the treated virus stocks with 30 mM imidazole to reduce unspecific binding (Figure R2B). This was even more pronounced for HIV-1, resulting in ~ 100-fold lower signals, precluding analysis. We thus prefer to keep the existing datasets, where PSDC was not removed but diluted.

11Figure R2: PSDC removal post-digestion does not substantially alter direct anti-ZIKV activity. ZIKV (MR766, diluted to achieve an MOI of 0.5 on cells) was treated with indicated concentrations of PSDC (µg/ml) for 30 min at 30°C. Imidazole was then spiked to 30 mM concentration in both samples to enable removal of PSDC while reducing unspecific binding to Ni-NTA beads and supernatants

either kept at RT (“not removed”) or PSDC actively removed by two rounds of HisPur Ni-NTA bead binding with magnetic depletion of bound PSDC. For removal, virus was incubated with beads for 1 h while rotating at RT in two subsequent depletion steps, removing beads by a tube magnet holder. Infection rates were determined by 4G2 immunodetection of ZIKV-E as described. (A) Normalized values of 3 independent experiments, each in triplicates (means ± SEM). (B) Raw value of representative experiment in triplicates (means ± SEM), dashed line indicates OD achieved with uninfected cells.

4. Fig. 2. What is critical concentration or topography of PS need to be acquired for interference with virion attachment?

We now show that PS concentrations as low as 0.02 mol% are sufficient for antiviral activity (new Figures 1C, 1D lines 112-114). Concerning topology, we hypothesized that multivalency i.e. the binding of multiple PS molecules on one vesicle to several PS receptors might enhance antiviral activity. This requires lateral mobility of PS molecules in the vesicular bilayer especially at lower PS concentrations. We investigated this by preparing liposomes containing 0.2 and 1 mol% PS in the background of 18:1 PC (DOPC, which has a sub-zero transition temperature and is thus liquid-ordered at 37°C) and 18:0 PC (DSPC, which has a transition temperature of 55°C and is thus expected to be more rigid at 37°C, impeding lateral mobility of PS). Indeed, rigid liposomes were ~10-fold less potent in inhibiting infection than liquid-ordered liposomes (new Extended Data Figure 2D, lines 114-118). Inclusion of cholesterol, another factor influencing membrane fluidity, attenuated this effect, although non-significantly. These data support our hypothesis that lateral mobility of PS to enable multivalent binding to PS receptors and contribute to antiviral potency.

5. A symmetry PS in EVs is not absolute. Decarboxylation of PS can lead to an exposure of luminal PS molecules from the inner to the outer leaflet of EVs. After removal of the PSD, a flop of PS from the inner to the outer leaflet would reestablish PS asymmetry. It is important to verify the membrane asymmetry of the isolated vesicles and to know how long it is retained.

This is an interesting point. To control for spontaneous flopping, we incubated a PS-shaved and blocked sample for 1 week at 4°C and another day at 37°C (new Figure 4F-I, lines 273-283). We found that low PS surface levels and reduced antiviral activity was maintained after prolonged incubation, suggesting that spontaneous PS translocation is either absent or a rare event. This agrees with data from liposomes, where PS-depleted liposomes maintained asymmetry for days at room temperature (Drechsler et al., Biophys J., 2018).

6. Figure 3H. A targeted bead-assisted flow cytometry was used identify and quantitate PS positive and negative sub-populations of EVs. It would be very interesting to know the exact percentage of EVs that presumably expose PS. Such information can be obtained from the binding of surface exposed PS to Ca⁺⁺-independent fluorophore-conjugated lactadherin

13lactadherin-A647. Representative raw data with flow cytometry quadrants or histograms should be provided in order to support a heat map shown on Fig. 3H (ESSENTIAL).

To analyze the percentage of individual EVs exposing PS, we applied a nanoscale flow cytometer (nanoFCM) and stained EVs for the tetraspanin CD9/CD63/CD81 markers as well as PS exposure by lactadherin A647. We found that a mean ~ 18% of TSPN-positive semen EVs expose PS (new Fig 3I; flow cytometry quadrants are shown in new Extended Data Fig. 3F; lines 215-218). Concerning the analysis of EVs by bead-based flow cytometry using the well-established commercially available product MACSPlex Exosome Kit, human (Miltenyi Biotec), we provide this representative dataset (Figure R3) and include a relevant citation which describes this method (line 899-900; Wikilander et al., *Frontiers Immunology*, 2019).

Figure R3: Representative raw data of bead-assisted flow cytometry for detection of EV-exposed PS using fluorescently tagged lactadherin. Single beads are first gated by SSC/FSC-A (top left) and individual beads containing antibodies directed against typical EV surface markers then separated by their endogenous PE/FITC-signal, enabling discrimination of 39 bead populations (37 markers + 2 isotype controls, large graph). The mean R660-APC/Alexa Fluro 647 fluorescence intensity of each bead population is then analyzed to reveal the proportion of PS-containing vesicles which simultaneously carry the protein marker of respective beads.

157. Cyclodextrin-mediated phospholipid exchange between PS liposomes and EVs can be utilized to supply PS back to EVs after PSDC treatment in order to keep the same protein-lipid background of different EVs to see whether these EVs retain the essentially the same ability to compete for viral apoptotic mimicry cell attachment and prevent an infection. (Li et al., Replacing plasma membrane outer leaflet lipids with exogenous lipid without damaging membrane integrity. PLoS One. 2019 Oct 7;14(10):e0223572

We appreciate the reviewer's suggestion of using cyclodextrins to modify the lipid envelope. Using PS-loaded methyl- α -cyclodextrin, we were able to resupply PS to PS-depleted EVs, further confirming the PS-dependent mode of action (new Figure 4F-I, lines 279-281).

Reviewer #2: The work of Groß et al. reports on a new function of extracellular vesicles (EVs): Defending against infection with a wide group of viruses that carry PS and use PS receptor for interaction with target cells. These viruses include Zika, Dengue, West Nile, Chikungunya, Ebola, and VSV, but not HIV-1, SARS-Cov-2, HCV, HSV and several other viruses. According to the authors, EVs that carry PS are generated upon viral infection and compete with viruses for binding to the cell membrane PS. Since the number of such EVs are far larger than the number of virions, viral infection is suppressed. The authors provide convincing experimental data that support this concept.

In particular, they showed that liposomes containing PS suppress Zika infection, while liposomes that contain only PC do not. Also, PS-containing liposomes do not suppress infection by HIV or HSV that do not use PS receptors for infection. To inhibit viral infection PS-containing liposomes have to be present before or during viral inoculation indicating that they interfere with viral entry. Confocal images revealed tat PS-containing liposomes are attached to the cell surface.

To bring their experimental system closer to the in vivo situation, the authors isolated EVs from pooled human semen, saliva, urine, breast milk, and blood. The amount of PS in these EVs varied depending on the body fluid they were isolated from. The highest concentration of PS was in semen, saliva, and urea, while the lowest was in blood. Significant amount of these EVs expose PS on their surface. Accordingly, hybrid EV-liposome vesicles and EVs from blood protect cells or cervical explants from ZIKV less efficiently than EVs from semen. There was no such effect when a panel of viruses that do not use PS receptors for viral entry. These results may explain the difference in viral infection pathways in nature.

In general, the results of the experiments seem to be convincing. Figures that accompany the text clearly support the conclusions. My critical remarks are minor and do not challenge the authors' concept.

We thank the reviewer for this positive evaluation.

1. It would be interesting to know what fraction of cell membrane PS are occupied by PS-

containing EVs at a particular concentration. This could have been evaluated in the course of the same experiments that the authors performed.

To address this, we stained for PS receptors Axl, presumed to be the major receptor of ZIKV, and measured the overlap of receptor and hybrid EV signals using z-stacks in confocal microscopy. Quantification of co-localisation of Axl and EVs saturated at a Mander's M2 coefficient of 0.91-0.92 at the highest concentrations, indicating that close to all Axl receptors are occupied by a Cy5-labeled EV (new Figure 4D+E). Extrapolating from this, an approximate 50% Axl occupation is reached at 2×10^{10} particles/ml, which is in the typical range of EV IC_{50} measured in antiviral assays (new Figure 4D, lines 263-269).

2. Another question is regarding the fate of the bound EVs. Using the mechanism of viral apoptotic mimicry, EVs should like viruses, enter the cells and decrease the amount of the surface-exposed PS. Do the authors have an idea of the timeline of this process and of the rate of the PS turnover?

One major difference between EVs and viruses engaging PS receptors is that viruses, upon endosomal uptake and acidification, will reveal a fusion peptide that enables endosomal escape and subsequent infection, while the fate of EVs is less clear. Most likely, at least a fraction of the receptor-bound EVs are taken up in a similar fashion as viruses. We confirmed this in a time-course experiment with Cy5-labeled hybrid semen EVs and LysoTracker where we observed co-localisation between EVs and lysosomes (new Extended Data Figure 5B+C, lines 269-271). Thus, endosomal uptake of EVs does certainly occur, but the exact kinetics and their fate need to be addressed in future studies.

3. How do the authors exclude direct interactions of EVs with virions that may lead to inactivation of the latter?

We have previously shown that EVs from semen and saliva interfere with virus attachment without targeting the virions (Müller et al., Nat Commun. 2018; Conzelmann et al., JEV, 2020). To corroborate this, we determined viral titers after incubating virions with semen EVs and PC or PS liposomes. Neither EVs nor liposomes reduced the infectious titer supporting that EVs do not directly inactivate ZIKV (new Extended Data Figure 4B, lines 253-254).

4. While the authors used hybrids of fluorescent liposomes and natural EVs to visualize these vesicles binding, why did not they use fluorescent dyes like Bodipy or similar to stain natural EVs isolated from body fluids?

Lipophilic dyes based on Bodipy, PKH, or similar compounds are a widely used option for staining EVs. However, these dyes are also known to cause issues that confound results in attachment assays (Dominkuš et al., BBA Biomembranes, 2018; Tekov et al., JEV, 2017). Even after removal of free dye by SEC, lipophilic dyes may stain cellular membranes in co-incubation experiments, again making interpretation difficult. We thus chose to use covalently labeled synthetic lipids, integrating them into the biological membranes of isolated EVs. This protocol is routinely applied for drug loading of EVs as it allows convenient analysis of the background signal caused by the labeling-liposomes.

5. Finally, when the authors present “representative” images, they should mention how many replicates these images in the Figures represent.

We have added this information to all Figure legends. In all experiments a minimum of 3 images per condition was analyzed.

Reviewer #3: The study by Grob et al. is a follow-up of a previous work published by the Muller’s team showing that semen and saliva contain extracellular vesicles (EVs) block Zika virus (ZIKV) infection both in vitro and in an ex-vivo model. Here, the authors have investigated the underlying mechanisms and showed that EVs from body fluids expose the lipid phosphatidylserine (PS) on their outer membrane layer. PS is also known to be exposed at the viral envelope of several viruses and is essential for their infectious entry. This mechanism, known as viral apoptotic mimicry, is inhibited by molecules that bind to PS with high affinity such as annexin A5.

The present study is indeed based on our previous research, which had implicated EVs in semen and saliva as inhibitors of ZIKV infection and offered a potential explanation for the scarcity of transmission events involving these fluids despite shedding of infectious virus into

them. Here, we defined the underlying mechanism and show that it has implications far beyond ZIKV, as affects all viral pathogens using apoptotic mimicry.

In this manuscript, the authors showed that PS-exposing EVs prevent infection of ZIKV and other enveloped viruses that expose PS at their surface such as dengue, chikungunya or Ebola viruses. In my opinion, this manuscript does not bring much new information when compared to what is already known in the literature.

The experiments described in the Figure 1, 2 and 5 of the manuscript are very well-performed but unfortunately most of the results presented in these figures are expected and a repetition of data already published by others (PMID: 21501828, PMID: 26052667, PMID: 23555248, PMID: 23084921).

The concept of viral apoptotic mimicry is indeed well established. For ZIKV specifically, viral apoptotic mimicry had been suggested based on closely related flaviviruses, but not yet demonstrated conclusively. As our discovery of EV-virion competition was based on ZIKV, we verified its PS-dependence experimentally. Similar verification experiments have been performed in studies examining other viruses including those cited by the reviewer. Our additional approach to modify the lipid envelope of virions goes beyond the state of the art. Using ZIKV as mimicry model enabled us to elucidate how abundant endogenous EVs in body fluids outcompete viruses by occupying PS receptors, with potential implications for the transmissibility of all viruses applying the strategy of viral apoptotic mimicry.

The novelty of this manuscript is the idea that PS-exposing EVs in secreted fluids may act as an innate defense barrier that regulate the transmission routes of viruses that use apoptotic mimicry. Unfortunately, the data presented by the authors do not support this concept.

Our concept convinced the other two reviewers and is supported by several lines of evidence. For example, epidemiological evidence shows that viruses using PS-receptors are rarely transmitted sexually or orally, although they are shed into these body fluids in high concentrations (Table S1). In addition, EVs purified from seminal plasma potently inhibit ZIKV infection of *ex vivo* human vaginal tissue and primary foreskin fibroblasts, representing initial target cells during sexual ZIKV transmission (Figure 5C). Together with our detailed elucidation

of the underlying molecular mechanism the hypothesis that EVs restrict sexual and oral transmission of PS-receptor using viruses seems highly warranted.

Overall, given the experimental redundancy of this manuscript with already published papers, this study lacks novelty and does not provide a significant advance in our understanding of the role of EVs in modulating the transmission route of apoptotic mimicry viruses. In my opinion, this manuscript is more suitable to a specialized journal.

We disagree. Our data strongly support a role of PS-exposing endogenous EVs in limiting human-to-human transmission of viral apoptotic mimicry-using viruses. Saturating PS receptors to prevent virion attachment is a powerful strategy of the innate immune system, which may have evolved in response to viral exploitation of PS receptors. Our discovery reveals a fascinating and highly relevant aspect of innate immunity and functional EV biology.

Decision Letter, first revision:

Message: Our ref: NMICROBIOL-23030622A

4th January 2024

Dear Janis,

Thank you for your patience as we've prepared the guidelines for final submission of your Nature Microbiology manuscript, "Phosphatidylserine-exposing extracellular vesicles in body fluids inhibit viral apoptotic mimicry infection" (NMICROBIOL-23030622A). Please carefully follow the step-by-step instructions provided in the attached file, and add a response in each row of the table to indicate the changes that you have made. Please also check and comment on any additional marked-up edits we have proposed within the text. Ensuring that each point is addressed will help to ensure that your revised manuscript can be swiftly handed over to our production team.

When you upload your final materials, please include a point-by-point response to any

21remaining reviewer comments.

In recognition of the time and expertise our reviewers provide to Nature Microbiology's editorial process, we would like to formally acknowledge their contribution to the external peer review of your manuscript entitled "Phosphatidylserine-exposing extracellular vesicles in body fluids inhibit viral apoptotic mimicry infection". For those reviewers who give their assent, we will be publishing their names alongside the published article.

Nature Microbiology offers a Transparent Peer Review option for new original research manuscripts submitted after December 1st, 2019. As part of this initiative, we encourage our authors to support increased transparency into the peer review process by agreeing to have the reviewer comments, author rebuttal letters, and editorial decision letters published as a Supplementary item. When you submit your final files please clearly state in your cover letter whether or not you would like to participate in this initiative. Please note that failure to state your preference will result in delays in accepting your manuscript for publication.

Cover suggestions

COVER ARTWORK: We welcome submissions of artwork for consideration for our cover. For more information, please see our guide for cover artwork.

Nature Microbiology has now transitioned to a unified Rights Collection system which will allow our Author Services team to quickly and easily collect the rights and permissions required to publish your work. Approximately 10 days after your paper is formally accepted, you will receive an email in providing you with a link to complete the grant of rights. If your paper is eligible for Open Access, our Author Services team will also be in touch regarding any additional information that may be required to arrange payment for your article.

Please note that *Nature Microbiology* is a Transformative Journal (TJ). Authors may publish their research with us through the traditional subscription access route or make their paper immediately open access through payment of an article-processing charge (APC). Authors will not be required to make a final decision about access to their article until it has been accepted. Find out more about Transformative Journals

Authors may need to take specific actions to achieve compliance with funder and institutional open access mandates. If your research is supported by a funder that requires immediate open access (e.g. according to Plan S principles) then you should select

22the gold OA route, and we will direct you to the compliant route where possible. For authors selecting the subscription publication route, the journal's standard licensing terms will need to be accepted, including self-archiving policies. Those licensing terms will supersede any other terms that the author or any third party may assert apply to any version of the manuscript.

Best wishes,

Reviewer #1:

Remarks to the Author:

I am fully satisfied with near exemplary responses (answered to my main concerns experimentally !) to my questions as such as the results of all new control experiments that eliminate my concern 2 (new experiments resulted in new Extended Data Figure 1A and clever new experiments with lactadherin successfully used to "silence" the antiviral activity of EVs as shown on new Figure 4F-I), concern 4 (new experiments resulted in new Figures 1C and D + new experiments which performed to address Authors hypothesis (new Extended Data Figure 2D), concern 5 (new Figure 4F-I), concern 6 (new Fig 3I and new Extended Data Fig. 3F) , concern 7 (new experiment resulted in new Figure 4F-I) as such as provided answers on my concern 3 (plus new experiments thoughtfully extended by Authors as shown in Fig.R2 for "reviewers eyes only") and other small modifications which strengthen the manuscript.

Reviewer #2:

None

Final Decision Letter:

Message 14th February 2024

:
Dear Janis,

23I am pleased to accept your Article "Phosphatidylserine-exposing extracellular vesicles in body fluids are an innate defense against apoptotic mimicry viral pathogens" for publication in Nature Microbiology. Thank you for having chosen to submit your work to us and many congratulations.

You may wish to make your media relations office aware of your accepted publication, in case they consider it appropriate to organize some internal or external publicity. Once your paper has been scheduled you will receive an email confirming the publication details. This is normally 3-4 working days in advance of publication. If you need additional notice of the date and time of publication, please let the production team know when you receive the proof of your article to ensure there is sufficient time to coordinate. Further information on our embargo policies can be found here:

<https://www.nature.com/authors/policies/embargo.html>

Please note that *Nature Microbiology* is a Transformative Journal (TJ). Authors may publish their research with us through the traditional subscription access route or make their paper immediately open access through payment of an article-processing charge (APC). Authors will not be required to make a final decision about access to their article until it has been accepted. Find out more about Transformative Journals

Authors may need to take specific actions to achieve compliance with funder and institutional open access mandates. If your research is supported by a funder that requires immediate open access (e.g. according to Plan S principles) then you should

select the gold OA route, and we will direct you to the compliant route where possible. For authors selecting the subscription publication route, the journal's standard licensing terms will need to be accepted, including self-archiving policies. Those licensing terms will supersede any other terms that the author or any third party may assert apply to any version of the manuscript.

Congratulations to you and your co-authors again! I'm looking forward to seeing your paper published.

All the best,